# Relation Between the Interannual Variability in the Stratospheric Rossby Wave Forcing and Zonal Mean Fields Suggesting an Interhemispheric Link in the Stratosphere

Yuki Matsushita[1], Daiki Kado[2], Masashi Kohma[1], and Kaoru Sato[1]

[1]Department of Earth and Planetary Science, the University of Tokyo, Tokyo, 113-0033, Japan
[2]Research Center for Advanced Science and Technology, the University of Tokyo, Tokyo, 153-8904, Japan

*Correspondence to*: Yuki Matsushita (matsushitayu@eps.s.u-tokyo.ac.jp)

**Abstract.** Focusing on the interannual variabilities in the zonal mean fields and Rossby wave forcing in austral winter, an interhemispheric coupling in the stratosphere is examined using reanalysis data: the Modern-Era Retrospective Analysis for Research and Applications, version 2 (MERRA-2). In the present study, the Eliassen-Palm (EP) flux divergence averaged over the latitude and height regions of 50°–30°S and 0.3–1 hPa, respectively, are used as a proxy of the Rossby wave forcing, where the absolute value of the EP flux divergence is maximized in the winter in the Southern Hemisphere (SH). The interannual variabilities in the zonal mean temperature and zonal wind are significantly correlated with the SH Rossby wave forcing in the stratosphere in both the SH and Northern Hemisphere (NH). The interannual variability in the strength of the poleward residual mean flow in the SH stratosphere is also correlated with the strength of the wave forcing. This correlation is significant even around the equator at an altitude of 40 km and at NH low latitudes of 20-40 km. The temperature anomaly is consistent with this residual mean flow anomaly. The relation between the cross-equatorial flow and the zonal mean absolute angular momentum gradient ($\bar{M}_y$) is examined in the meridional cross section. The $\bar{M}_y$ around the equator at the altitude of 40 km is small when the wave forcing is strong, which provides a pathway for the cross-equatorial residual mean flow. These results indicate that an interhemispheric coupling is present in the stratosphere through the meridional circulation modulated by the Rossby wave forcing.

## 1 Introduction

The Brewer-Dobson circulation (BDC) is composed of the residual mean circulation and isentropic mixing in the stratosphere. The deep branch of the BDC is mainly driven by planetary-scale Rossby waves on the winter hemisphere (Butchart, 2014). The maximum tropical upwelling is observed on the summer hemisphere of the equatorial region (e.g., Plumb and Eluszkiewics, 1999; Tung and Kinersley, 2001; Okamoto et al., 2011). However, the Rossby wave forcing in the winter extratropics does not directly drive the cross-equatorial flow around the equator since the wave forcing cannot be balanced with Coriolis force associated with meridional wind owing to small Coriolis parameter $f$. Although the meridional circulation in the extratropics requires wave forcing to cross angular momentum ($\bar{M}$) contours aligned nearly vertically, the meridional

circulation can exist around the equator without wave forcing because the $\overline{M}$ contours are horizontally aligned. (e.g., Plumb and Eluszkiewics, 1999). Tomikawa et al. (2008) used high-resolution general circulation model (GCM) simulation data to show that the strong residual mean flow crosses the equator along nearly horizontally aligned contours of $\overline{M}$.

Previous studies have highlighted the importance of Rossby waves in the stratospheric interannual variabilities from various viewpoints, such as extratropical zonal winds in the winter modulated by the location of the westerly maximum in the winter

in the SH (Shiotani et al., 1993; Kodera and Kuroda, 2002), the spring time temperature and dates of the stratospheric final warming in the SH (Newman et al., 2001; Black and McDaniel, 2007; Hirano et al., 2016), and the stratospheric quasi-biennial oscillation (QBO) in the Northern Hemisphere (NH) (Holton and Tan, 1980; Yamashita et al., 2011) and Southern Hemisphere (SH) (Baldwin and Dunkerton, 1998; Salby et al., 2011). Holton and Tan (1980) defined the QBO phase using monthly mean zonal wind at 50 hPa near the equator (at Balboa, Canal Zone, 9°N) and showed that the amplitude of wavenumber 1 Rossby

waves in the northern early winter is stronger during the easterly phase of the QBO. Baldwin and Dunkerton (1998) and Salby et al. (2011) showed the significant influence of the QBO phase on the southern polar vortex and Antarctic ozone anomaly. Young et al. (2011) showed the interannual variability in the stratospheric temperature in the tropical regions and winter extratropical regions with out-of-phase relation and noted a close relation with the Rossby waves interannual variability. Kodera and Kuroda (2002) proposed a possible dynamical impact of the 11-year solar cycle on the winter hemisphere using

reanalysis data over 20 years from 1979–1998. They suggested that the solar cycle controls the transition period from a radiatively controlled state of the circulation around the stratosphere in the winter hemisphere to a dynamically controlled state and proposed a mechanism connecting the westerly jet around the stratopause to the 11-year solar cycle through the interaction with Rossby waves.

Stationary planetary-scale Rossby waves have large amplitudes in the winter stratosphere in both hemispheres (Randel, 1988).

Since stationary Rossby waves can propagate vertically in weak westerlies (Charney and Drazin, 1961), it has been considered that the direct effect of stationary Rossby waves is confined to the winter hemisphere. Recent studies suggest the presence of the interhemispheric coupling between the winter stratosphere and summer upper mesosphere through modulation of the mesospheric circulation, which is mainly driven by gravity waves and is initiated by the stationary Rossby wave forcing in the winter hemisphere (e.g., Becker et al., 2004; Körnich and Becker, 2010).

In the present study, the relation of the interannual variability in the Rossby wave forcing in the SH winter to that of the zonal mean fields in both hemispheres is investigated using reanalysis data over 38 years. The magnitude of the Eliassen-Palm (EP) flux divergence is maximized around the winter extratropical stratopause, which is used here as a proxy for the Rossby wave forcing. We perform correlation analyses between the Rossby wave forcing and zonal mean fields in both hemispheres to clarify the dynamical coupling across the equator. The remainder of this paper is organized as follows. The data description

and method of the analysis are given in Sect. 2. The results of the analyses in terms of the Rossby wave forcing characteristics in austral winter, the correlation observed in the interannual variabilities, and the relation with the cross-equatorial flow are shown in Sect. 3. The signals in the mesosphere, the impacts of the solar forcing and QBO, and the results for an extended period and for the boreal winter are discussed in Sect. 4. A summary and concluding remarks are given in Sect. 5.

## 2 Data and Methods

In this study, the Modern-Era Retrospective Analysis for Research and Applications, version 2 (MERRA-2, Geralo et al., 2017) is used. The interannual variability in the zonal mean fields averaged from June to August (JJA) of 1980–2017 is the focus period in this study. We analysed JJA-averaged fields to reduce the transient feature. We also confirmed that the results for the respective month are qualitatively similar to those for JJA-averaged fields. The residual mean flow $(\bar{v}^*, \bar{w}^*)$, zonal mean absolute angular momentum $\bar{M}$, and the EP flux $\boldsymbol{F} = \left(0, F^{(\phi)}, F^{(z)}\right)$ are calculated using the MERRA-2 dataset:

$$\bar{v}^* \equiv \bar{v} - \rho_0^{-1}(\rho_0\overline{v'\theta'}/\bar{\theta}_z)_z,$$

$$\overline{w}^* \equiv \overline{w} + (a\cos\phi)^{-1}(\cos\phi\overline{v'\theta'}/\bar{\theta}_z)_\phi,$$

$$\bar{M} \equiv (z+a)(\bar{u}\cos\phi + (z+a)\Omega\cos^2\phi),$$

$$F^{(\phi)} \equiv \rho_0 a\cos\phi(u_z\overline{v'\theta'}/\bar{\theta}_z - \overline{v'u'}),$$

$$F^{(z)} \equiv \rho_0 a\cos\phi\left([f - (a\cos\phi)^{-1}(\bar{u}\cos\phi)_\phi]\overline{v'\theta'}/\bar{\theta}_z - \overline{w'u'}\right),$$

where the overbar and prime denote the zonal mean and deviation from the zonal mean, respectively; $u$, $v$, and $w$ are the zonal, meridional, and vertical wind components, respectively; $f$ is the Coriolis parameter; $\theta$ is the potential temperature; $\phi$ is the latitude; $a$ is the mean radius of the Earth; $\Omega$ is the rotation rate of the Earth; and $\rho_0$ is the basic density. The subscripts denote partial derivatives. The EP flux is commonly used to quantitatively diagnose Rossby wave activity. The wave forcing term in the zonal momentum equation of the transformed Eulerian-mean (TEM) system (Andrews and McIntyre, 1976) is written as the divergence of the EP flux:

$$\frac{\partial\bar{u}}{\partial t} + \bar{v}^*\left[\frac{(\bar{u}\cos\phi)_\phi}{a\cos\phi} - f\right] + \overline{w}^*\bar{u}_z = \frac{1}{\rho_0 a\cos\phi}\nabla\cdot\boldsymbol{F} + \bar{X},$$

where $\bar{X}$ is the other forcing term, such as that due to gravity waves.

## 3 Results

First, the interannual variability in the zonal mean fields in JJA is examined. Figures 1a and 1b show the meridional cross-section of the standard deviation of $\bar{u}$ and $\bar{T}$, respectively, over 38 years from 1980–2017. Large interannual variabilities are observed for both $\bar{u}$ and $\bar{T}$ in the SH extratropical region (90°–20°S) above a height of 30 km. There are also large interannual variabilities around the equator in the height range of 20–40 km, which is likely to be associated with the quasi-biennial oscillation.

Figure 1c shows the climatology of three-hourly values of the root mean square of the geopotential height deviation ($Z'$) from the zonal mean obtained at every three hours (namely, $\sqrt{\overline{Z'^2}}$), which roughly corresponds to the climatological amplitude of Rossby waves and is referred to as the wave amplitude. Large wave amplitudes are observed in the SH extratropical region, with a maximum at approximately 60°S in the height range of 45–50 km. In the NH, the wave amplitudes are small compared with those in the SH. This difference is likely because the easterly winds in the summer middle atmosphere prevent the upward

propagation of stationary Rossby waves (Charney and Drazin, 1961). Figure 1d shows the climatology of EP flux divided by $\rho_0 a \cos\phi$ (arrows) and its divergence (colours). Upward propagating waves from the troposphere to the stratosphere are observed around 60°S. The waves are refracted towards the equator with height. The westward wave forcing is strong above 30 km in the SH extratropical region. The wave forcing maximum is located in the latitude and height region of approximately 50–55 km (0.3–1 hPa), 50°–30°S, which is hereinafter referred to as Region A.

In the following, a correlation analysis is performed using the EP flux divergence averaged over Region A in JJA (denoted as $[\nabla \cdot \boldsymbol{F}]_A$) for each year as a proxy for the Rossby wave forcing in the SH winter. Note that $[\nabla \cdot \boldsymbol{F}]_A$ decreases as the Rossby wave forcing increases because the breaking and/or dissipation of upward propagating Rossby waves results in a westward (i.e., negative) forcing. Thus, the correlation sign of the interannual variability in the zonal mean fields to $[\nabla \cdot \boldsymbol{F}]_A$ is opposite to that of the corresponding anomaly.

Figure 2a shows the correlation between the interannual variability of $[\nabla \cdot \boldsymbol{F}]_A$ and that of $\sqrt{\overline{Z'^2}}$ in the meridional cross-section. The regions with statistically significant correlations at a confidence level higher than 95% are coloured. A high correlation is observed in the SH extratropical region, where $\sqrt{\overline{Z'^2}}$ is climatologically large (Fig. 1a). In the maximum region of the wave amplitude observed at approximately 60°S latitude and 45 km altitude, the absolute values of the correlation are higher than 0.8. Figure 2b shows the correlation between the interannual variabilities of $[\nabla \cdot \boldsymbol{F}]_A$ and those of $(\rho_0 a \cos\phi)^{-1}\nabla \cdot \boldsymbol{F}$. The significant correlation is not confined in Region A but is widely extended to ~10°–70°S and ~35–60 km. Thus, $[\nabla \cdot \boldsymbol{F}]_A$ represents not only the interannual variability of Rossby wave forcing in Region A but also that in a wider region around Region A.

Another EP flux convergence maximum is observed in the high latitude region of 90°–80°S above 50 km (Figure 1d). However, the correlation between $\sqrt{\overline{Z'^2}}$ and the wave forcing in this region was not significant in the meridional cross-section of the winter stratosphere (not shown). It seems that this maximum is not related to the interannual variability in the Rossby waves propagating upward through the stratosphere. Thus, further analysis regarding this wave forcing in the polar region is not performed.

Figure 3a shows the correlation between the interannual variabilities in $[\nabla \cdot \boldsymbol{F}]_A$ and $\bar{T}$. Below Region A, the correlation is significantly negative at 70°–30°S and positive at 30°S–30°N. Notably, the positive correlation peaks in both hemispheres of the equatorial region, in the latitude and height regions of ~10°–20° and ~30–40 km. The significant correlation exhibits a Π-shaped spatial pattern. The correlation coefficients of the positive peaks in the SH and in the NH are comparable. This indicates that the wave forcing in the SH is related to the mean fields in the low latitude region of the NH. In addition, opposite-sign correlations are seen above Region A; positive correlations are observed at 70°–30°S and negative correlations are observed at 30°S–30°N. The significant correlation shows a quadrupole structure that extends to the NH around Region A.

Here, the characteristics of the reanalysis dataset used in the present study should be addressed. Gelaro et al. (2017) noted that the globally averaged $\bar{T}$ in MERRA-2 changes discontinuously when new observational data are introduced into the data assimilation process. They showed that the discontinuous change is not large in the lower stratosphere but is more obvious in

the upper stratosphere and mesosphere. A remarkable temperature change is observed around the stratopause before and after 2004, when the Earth Observation System Aura Microwave Limb Sounder (MLS) data were introduced for assimilation. To assess the effect of this reanalysis data discontinuity on the present results, the same analyses as those shown in Fig. 3a are

conducted but separately for 1980–2004 and 2005–2017 (Figs. 3c and 3d, respectively). The spatial patterns of correlations, such as the quadrupole pattern around Region A and the Π-shaped pattern in the equatorial stratosphere, are still observed for both time periods. This indicates that the change in $\bar{T}$ due to the introduction of MLS temperature has a minor impact on the interannual variability related to the Rossby wave forcing in the reanalysis dataset.

Figure 3b shows the correlation between the interannual variability in $[\nabla \cdot \boldsymbol{F}]_A$ and that in $\bar{u}$. In the SH, the correlation is

significantly negative at 80°–50°S and positive at 50°–20°S above 20 km. Significant correlations are also observed in the NH (0°–60°N and 35–60 km). This finding indicates that the strong wave forcing in Region A is related to a weak (strong) westerly in the SH low (high) latitude region and a strong easterly in the NH low and middle latitude regions. The distribution of the correlation coefficient of $\bar{u}$ is qualitatively consistent with that of $\bar{T}$ in terms of the thermal wind balance.

The correlation coefficients for June, July, and August are also calculated separately. Although the correlation coefficient

becomes smaller than result for the JJA mean, it is confirmed that the spatial patterns of the correlation for $\bar{T}$ and $\bar{u}$ for each month are qualitatively similar to Figure 3a and 3b, respectively (not shown).

The correlation of the residual mean flows $(\bar{v}^*, \bar{w}^*)$ with $[\nabla \cdot \boldsymbol{F}]_A$ is shown in Figs. 4b and 4a. Below Region A, the correlation of $\bar{w}^*$ is significantly positive at 70°–30°S, as shown in Fig. 4a. This means that $\bar{w}^*$ is downward when the wave forcing in Region A is strong. In Fig. 4b, a significantly positive correlation of $\bar{v}^*$ is observed in the latitude and height regions

of 20°–60°S and 35–55 km, corresponding to the region in which the correlation of the EP flux divergence with $[\nabla \cdot \boldsymbol{F}]_A$ is significant (Fig. 2b). This finding indicates that $\bar{v}^*$ is poleward where the wave forcing is strong. Interestingly, there are also significantly positive correlations of $\bar{v}^*$ at 30°S–30°N and at 35–45 km and 50–60 km. In particular, the positive correlation is high below 40 km in the NH. These spatial patterns are also consistent with the correlation of $\bar{T}$ (Fig. 3a) through the adiabatic processes associated with vertical motions. The strong wave forcing maintains a downwelling and high temperature at

approximately 70°–30°S, cross-equatorial southward circulation at approximately 40 km, and upwelling and low temperatures at approximately 20°S–30°N. In fact, the characteristic Π-shaped structure in the $\bar{T}$ correlation (Fig. 3a) is also seen in $\bar{w}^*$, although the spatial pattern of the correlation of $\bar{w}^*$ slopes down towards the north.

If the meridional gradient of $\bar{M}$ $(\bar{M}_y)$ is nonzero, the wave forcing is necessary to maintain the meridional circulation (Plumb and Eluszkiewicz, 1999). At low latitudes, $\bar{M}_y$ can be zero with background zonal wind shear, which permits meridional

movement of air parcels even in the absence of a wave forcing (e.g., Tomikawa et al., 2008). Such an equatorial meridional flow exists along the $\bar{M}$ contour to satisfy the mass continuity with extratropical wave-driven circulation in the winter hemisphere. A climatological meridional cross-section of $\bar{M}$ and $\bar{M}_y$ in JJA is shown in Figure 5a. At the equator, $\bar{M}$ has a minimum at 25–30 km and maxima at ~15 km and ~55 km. In the tropics, $\bar{M}_y$ is generally small compared with that at other latitudes.

To examine the interannual variabilities in $\bar{M}$, a composite analysis is performed with respect to the anomalies of $[\nabla \cdot \boldsymbol{F}]_A$ from the climatology. The strong (weak) wave forcing years are defined as the years with $[\nabla \cdot \boldsymbol{F}]_A$ anomalies, which are smaller (larger) than $-0.5\sigma$ ($0.5\sigma$), where $\sigma$ is the standard deviation of $[\nabla \cdot \boldsymbol{F}]_A$ (Fig. 5d). As a result, 13 years are chosen as the strong wave forcing years (1985, 1988, 1992, 1993, 1996, 1997, 2002, 2004, 2005, 2010, 2012, 2013, and 2017) and 15 years are chosen as the weak wave forcing years (1980, 1981, 1983, 1987, 1989, 1995, 1998, 1999, 2000, 2001, 2003, 2008, 2009,

2011, and 2015). Figures 5b and 5c show the composite of $\bar{M}$ and $\bar{M}_y$ for the strong and weak wave forcing years, respectively. The absolute angular momentum around the equator at 35–40 km is small (large) when the wave forcing is strong (weak). At these altitudes, the region of small $\left|\bar{M}_y\right|$ in the strong wave forcing years extends to higher latitudes. The altitudes of these variabilities are in accordance with that of the cross-equatorial $\bar{v}^*$ that is correlated with $[\nabla \cdot \boldsymbol{F}]_A$ in Figure 4b. This relation between $\bar{M}$ and $[\nabla \cdot \boldsymbol{F}]_A$ is also observed in Fig. 3b. The correlation of $\bar{u}$ with $[\nabla \cdot \boldsymbol{F}]_A$ is significantly positive around 40 km

10°S–10°N, which is consistent with the result of the composite analyses.

    To confirm the relation between $\bar{M}_y$ and $\bar{v}^*$ around the equator, we define the region of 10°S–10°N, 35–45 km as Region B, and examine $\bar{M}$ averaged over Region B (hereafter referred to as $[\bar{M}]_B$). Note that small values of $[\bar{M}]_B$ correspond to small values of equatorial $\bar{M}_y$ because $\bar{M}$ reach a latitudinal maximum around the equator. The correlation between $[\nabla \cdot \boldsymbol{F}]_A$ and $[\bar{M}]_B$ is significantly positive (0.49), which is consistent with the results of the composite analyses. The correlation between

the interannual variability of $[\bar{M}]_B$ and $\bar{v}^*$ is shown in Fig. 6. The correlation is high and significantly positive in the region of the cross-equatorial flow indicated in Fig. 4b, at 60°S–50°N and 35–45 km. Thus, when the absolute angular momentum at the Region B is small, the southward cross-equatorial flow through the Region B is strong.

    Semeniuk and Shepherd (2001) examined the middle-atmosphere Hadley circulation and its interaction with extratropical wave-driven circulation, using a numerical model. They showed that the extratropical wave-driven circulation affects the $\bar{M}_y$

around the equator together with the middle-atmosphere Hadley circulation and that the significant overturning of $\bar{M}$ contours at the equator is attributable to the combination of the middle-atmosphere Hadley circulation and the extratropical wave-driven circulation. Thus, the wave forcing in the Region A is likely to modify the residual mean circulation in two ways: driving the residual mean flow in the SH and modifying the mean wind around the equator with a low $\left|\bar{M}_y\right|$.

    We performed the same composite analysis but separately for the periods of 1980–2004 and 2005–2017 to examine the

impact of the temperature discontinuity in MERRA-2 and confirmed that the results are qualitatively the same as those for 1980–2016 (not shown).

## 4 Discussion

### 4.1 Relation with previous studies

In Sect. 3, the correlation between the wave forcing in Region A ($[\nabla \cdot \boldsymbol{F}]_A$) and the zonal mean fields in the altitudes below Region A is shown. Specifically, the correlation between the interannual variabilities in $\bar{T}$ and $[\nabla \cdot \boldsymbol{F}]_A$ is significantly negative at 70°–30°S and positive at 30°S–30°N below Region A. In contrast, the opposite-sign correlation is observed above Region A, namely, there is a positive (negative) correlation at 70°–30°S (30°S–30°N), which forms a quadrupolar pattern in the correlation coefficient together with the correlation below Region A. It is known that the quadrupolar pattern of the temperature change appears when the stratospheric sudden warming occurs (Labitzke, 1972). Matsuno (1971) showed that the quadrupolar structure of temperature change can be interpreted as the transient response to the forcing of planetary waves during stratospheric sudden warming. However, the quadrupolar structure observed in the present study may not be fully explained by the transient response to the planetary waves since the present results are based on the JJA-averaged field. Instead, the observed quadrupolar pattern in the present study may be explained as follows. When the Rossby wave forcing around Region A is strong, the poleward flow anomaly around Region A and the downwelling anomaly at high latitudes below Region A is induced in the extratropical stratosphere from the downward control theory (Haynes et al., 1991). Since the $\bar{M}_y$ around the equator at ~40 km is small in the strong wave forcing years (Fig. 5), this poleward flow anomaly extends to low latitudes and crosses the equator at the altitude of ~40 km (Fig.6). And then, upwelling anomaly is formed at low latitudes due to the mass continuity. These vertical flow anomalies induce adiabatic heating, which corresponds to the lower part of the quadrupolar pattern. The zonal wind field is also modulated to maintain the thermal wind balance with the modulated temperature field. The westerly is weakened in the austral winter stratosphere, and the upward propagation of gravity waves with the westward phase speeds to the mesosphere is prevented. This modulation of the gravity wave propagation weakens the mesospheric meridional circulation, which induces the temperature anomalies adiabatically.

Körnich and Becker (2010), Karlsson and Becker (2016), and Gumbel and Karlsson (2011) showed the interhemispheric coupling in which the Rossby wave forcing in the winter hemisphere modifies the temperature around the summer polar mesopause and controls the interannual variability in the polar mesospheric clouds. This coupling is caused by the modulation of the mesospheric circulation driven by gravity waves where the propagation is affected by the mean wind change by the Rossby wave forcing in the winter hemisphere. The interhemispheric coupling shown in the present study is different from that shown in previous studies but occurs in the stratosphere only by the Rossby waves, and the gravity waves are not important in the mechanism.

### 4.2 On the impact of the external forcing

The relation of the interannual variabilities in the stratosphere with the solar cycle is examined in the present study and discussed in this section. Kodera and Kuroda (2002) noted that the seasonal evolution of the winter stratopause jet is considered

to be the transition from a radiatively controlled state, in which the wave forcing is small and the zonal wind is roughly determined by a radiative forcing, to a dynamically controlled state, when the wave amplitude is large and radiative forcing to
the zonal wind is small. They showed that the timing of this transition is largely affected by the 11-year solar cycle as well as the interannual variabilities in the wave forcing in the lower stratosphere. The proposed mechanism is as follows. During the solar maximum phase, the winter stratospheric jet remains in the radiatively controlled state for a longer period due to the enhanced meridional temperature gradient between the equatorial and polar regions (Kodera and Yamazaki, 1990). Since the winter stratospheric jet remains strong during a radiatively controlled state, planetary scale Rossby waves propagating to the
winter upper stratosphere are deflected from the midlatitudes to higher latitudes. The reduced Rossby wave forcing in the midlatitude region leads to the weakening of the meridional circulation and upwelling around the equator in the stratosphere. According to this mechanism, the stratospheric interhemispheric coupling examined in the present study may be attributed to the solar cycle.

The 10.7 cm solar radio flux (F10.7 index) averaged from June to August is used as the proxy for solar activity. Figure 7
shows the time series of $[\nabla \cdot \boldsymbol{F}]_A$ and the F10.7 index. The solar activity clearly exhibits an 11-year cycle oscillation. Note that the magnitude of the F10.7 index in the solar maximum phase obviously differs in each cycle, although the F10.7 index shows similar values in the solar minimum phases. The F10.7 index at the solar maximum phase at approximately 1990 is large and decreases in the later maximum phases. In contrast, the wave forcing shows a clear interannual variability with similar amplitudes during the displayed time period. In the time period before 2004, which overlaps with the period analysed in Kodera
and Kuroda (2002), $[\nabla \cdot \boldsymbol{F}]_A$ seems to be synchronized with the F10.7 index. The correlation is positive (0.41) but not statistically significant. In contrast, after 2004, $[\nabla \cdot \boldsymbol{F}]_A$ and the F10.7 index are roughly out of phase, although the correlation is not significant (-0.21). The correlation between $[\nabla \cdot \boldsymbol{F}]_A$ and F10.7 during the whole period is not statistically significant (0.29). The change in the relation between the 11-year solar cycle and atmosphere at ~2000 was also reported in Hervig et al. (2015). They noted that the response of polar mesospheric clouds (PMCs) to the 11-year solar cycle is obvious in the 1980s
and 1990s, while the PMCs response to the solar cycle is absent during 2000–2014. They discussed several possible explanations for this change observed at approximately 2000, which includes an apparent solar forcing amplification due to volcanic eruptions, namely, the eruptions of El Chichon in 1982 and Pinatubo in 1991.

The QBO can also modulate the extratropical circulation and the Rossby wave in the winter hemisphere. Following Salby et al. (2011), we use $\bar{u}$ at the equator and 30 hPa as a proxy of the QBO phase. Figure 8 shows the time series of $[\nabla \cdot \boldsymbol{F}]_A$ and
of $\bar{u}$ at the equator, 30 hPa. Although both time series show short-term variability, their typical periods of oscillation seem to be different, and the correlation is small and is not significant (-0.14). This result is consistent with that in Fig. 3b since the correlation between $[\nabla \cdot \boldsymbol{F}]_A$ and $\bar{u}$ is also small and not significant around the equator in the altitude range of 15-25 km. In the present analyses for the JJA-averaged fields, there is no significant relation between the Rossby wave forcing at Region A and the QBO on the interannual time scale. Since the reason why the correlation between the QBO and the wave forcing in
Region A is insignificant is out of the scope of this study, we only note here that the height region for the wave forcing in the

present study (namely, Region A) is located at a much higher altitude than that was focused in the previous studies (e.g., Baldwin and Dunkerton, 1998; Salby et al., 2011).

### 4.3 Results for zonal mean fields averaged over May through November and boreal winter

In the present study, we have examined the JJA-averaged field in terms of the interannual variability of the Rossby wave forcing and its relation to zonal mean fields. The same analysis is performed for the fields averaged over May through November, when the Rossby wave in the stratosphere are active in the SH. Although the results for the extended time period in the winter SH are quite similar to those for JJA, the correlation coefficients are weak, especially in the summer NH, and the statistically significant response of $\bar{u}$, $\bar{T}$, and $\bar{v}^*$ is limited only up to the equator. This is likely because the zonal mean fields and wave forcing in the NH in May and the SH in November are largely different (e.g., Randel, 1988). Furthermore, the
tendency of the zonal mean fields is not negligible in the zonal momentum equation for the equinoctial seasons. Because the seasonal evolution of the zonal mean fields is responsible for that of radiative heating, the temporal change of the zonal mean fields in JJA is in general small compared with the equinoctial season (Sato and Hirano, 2018), which is a preferable condition for the downward control principle analyses. As a result, the correlation analyses for JJA clearly indicate the interhemispheric link between the wave forcing in the winter SH and zonal mean fields in the summer NH.

Last but not the least, results for the interhemispheric link in the boreal winter are briefly described. To examine the interhemispheric link in the NH winter season, zonal mean fields averaged over December to February (DJF) of 1981–2017 are analysed. From a latitude-pressure section of the climatological wave forcing in DJF (not shown), it is seen that the wave forcing has a maximum in the region of 0.3–1 hPa and 30°–50°N, hereafter referred to as Region C. The correlation of DJF-averaged $\bar{u}$ with the wave forcing averaged over Region C ($[\nabla \cdot \boldsymbol{F}]_C$) is shown in Fig. 9a. Although the correlation is
significantly positive around Region C and in 30°S–30°N, 45–55 km, the correlation is not statistically significant at the latitudes higher than 30°S. Figure 9b shows the correlation of DJF-averaged $\bar{v}^*$ with $[\nabla \cdot \boldsymbol{F}]_C$. The correlation is significantly negative at Region C and at 10°S–30°N. It is indicated that the interhemispheric link and cross-equatorial flow in the boreal winter is associated with the wave forcing in Region C, while the latitudinal extent to the summer SH is limited compared to the austral winter (Figs. 3a and 4b). The difference in the correlation between JJA and DJF may be explained by the linearity
of the response of the zonal mean fields to the wave forcing. Due to large amplitude of planetary waves in the NH winter, which sometimes causes the breakdown of the polar vortex, a linear relation is unlikely obtained between the wave forcing and mean fields in the NH winter. The detailed analysis of the NH winter is beyond the scope of this paper.

### 5 Summary and concluding remarks

The relation of the interannual variabilities in the zonal mean fields and those of the Rossby wave forcing was examined
using MERRA-2 reanalysis data with a focus on the interhemispheric coupling through the stratosphere. In the SH in winter,

the Rossby wave forcing is maximized in the region of 30°–50°S and 0.3–1 hPa (Region A) and shows a large interannual variability. The interannual variability in the Rossby wave forcing in this region ($[\nabla \cdot \boldsymbol{F}]_A$) is correlated with that of other zonal mean fields, even in the NH stratosphere at low latitudes as well as in the SH stratosphere at nearly all latitudes. Specifically, a negative correlation of $\bar{T}$ with $[\nabla \cdot \boldsymbol{F}]_A$ at 30°S–30°N and a positive correlation of $\bar{u}$ with $[\nabla \cdot \boldsymbol{F}]_A$ at 50°S–60°N are

statistically significant. Correspondingly, a significant correlation of the residual mean flow with $[\nabla \cdot \boldsymbol{F}]_A$ is observed in both the NH and the SH. The spatial pattern of the correlation suggests that the material circulation crosses the equator at an altitude of ~40 km and that the NH part of this circulation is affected by the Rossby wave forcing in the SH Region A. To investigate the possible pathways of the cross-equatorial flow, we performed a composite analysis for absolute angular momentum ($\bar{M}$) by dividing it into strong and weak wave forcing years. The meridional gradient of $\bar{M}$ at ~40 km is small in the strong wave

forcing years when the cross-equatorial residual mean flow is observed.

The Rossby wave forcing in the SH stratosphere drives the residual mean circulation at the SH mid-latitudes, and this circulation connects the stratospheric zonal mean field of the SH and the NH over the equator. The cross-equatorial residual mean flow is not directly driven by the Rossby wave forcing but indirectly maintained by the mass-continuity and small meridional gradient of the absolute angular momentum around the equator.

Tomikawa et al. (2008) noted the interaction of the cross-equatorial flow around the stratopause and the semiannual oscillation of the zonal-mean zonal wind at the stratopause (S-SAO) on the seasonal time scale. The interannual variability in the S-SAO may play an important role in the interhemispheric coupling in the stratosphere. Details of the process of how the anomalies in the austral winter stratosphere propagate to the boreal summer stratosphere should be clarified in future studies.

The interhemispheric coupling shown in the present study occurs in the stratosphere through the modulation of the residual

mean circulation and absolute angular momentum field by the Rossby wave forcing. The mechanism is different from the interhemispheric coupling through the mesosphere caused by the modulation of gravity waves.

Data availability.

The MERRA-2 dataset is provided by NASA's Global Modelling and Assimilation Office (https://gmao.gsfc.nasa.gov/). The F10.7 index in the OMNI 2 dataset is used in the present study. The OMNI data were obtained from the GSFC/SPDF

OMNIWeb interface at https://omniweb.gsfc.nasa.gov.

Acknowledgements.

We greatly appreciate Kunihiko Kodera for his helpful comments and discussions. This work was supported by JST CREST JPMJCR1663. All figures in the present study are drawn using Dennou Club Library (DCL).

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

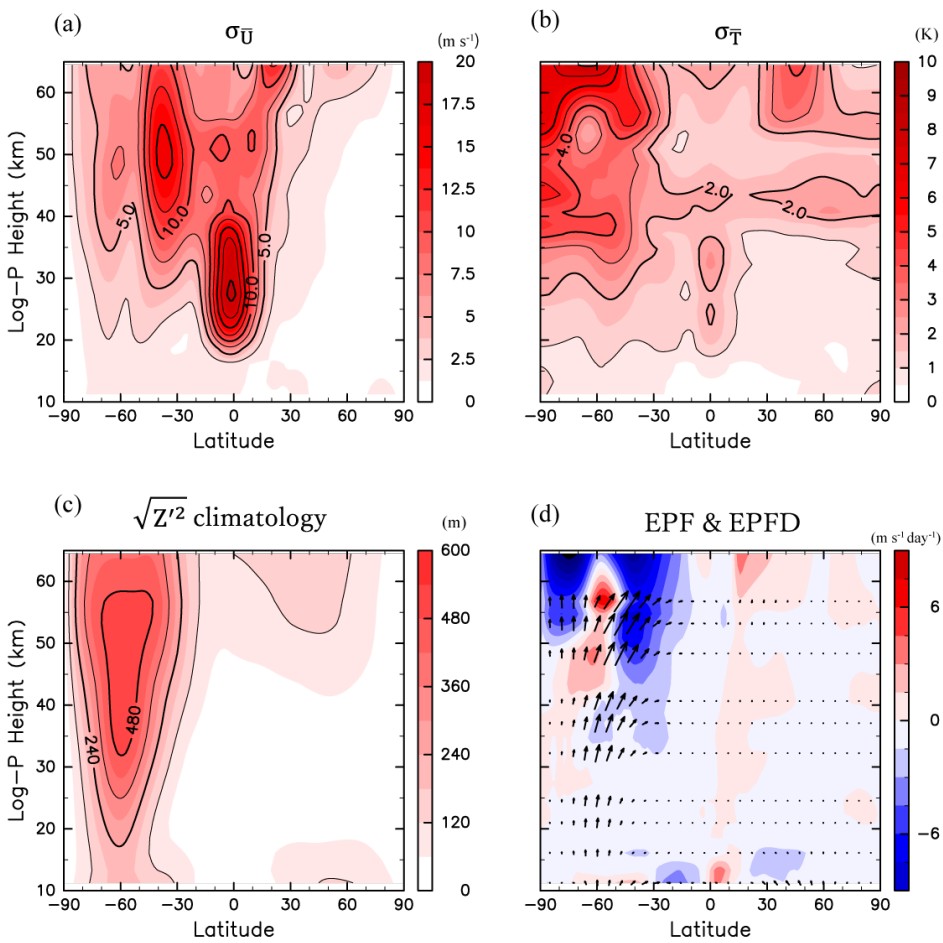

**Figure 1: Meridional cross-sections of the standard deviation of (a) $\bar{u}$ (m s⁻¹) and (b) $\bar{T}$ (K) from the climatology, climatology of (c) $\sqrt{\overline{Z'^2}}$ (m), and (d) EPF (vector) and EPFD (colour, m s⁻¹ day⁻¹); the contour intervals are (a) 1 m s⁻¹, (b) 2.5 K, and (c) 120 m, respectively.**


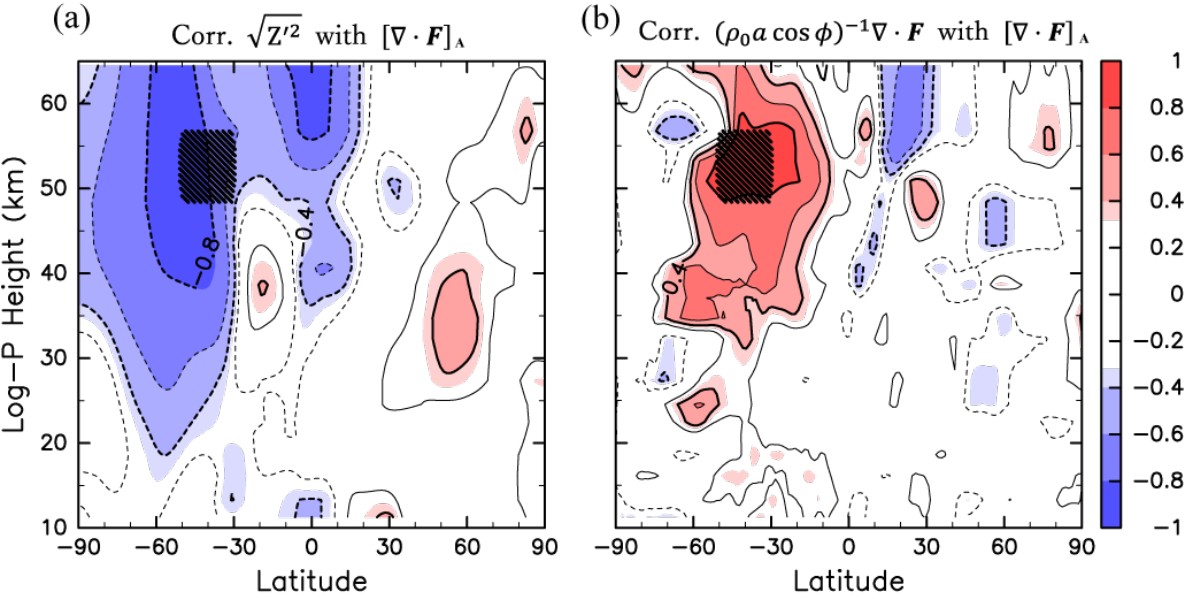

**Figure 2: Meridional cross-sections of the correlation coefficient of $[\nabla \cdot F]_A$ with (a) $\sqrt{\overline{Z'^2}}$ and (b) $(\rho_0 a \cos \phi)^{-1} \nabla \cdot F$ for JJA of 1980–2017. Contour intervals are 0.2, and the zero contours are omitted. Region A is indicated by hatching.**


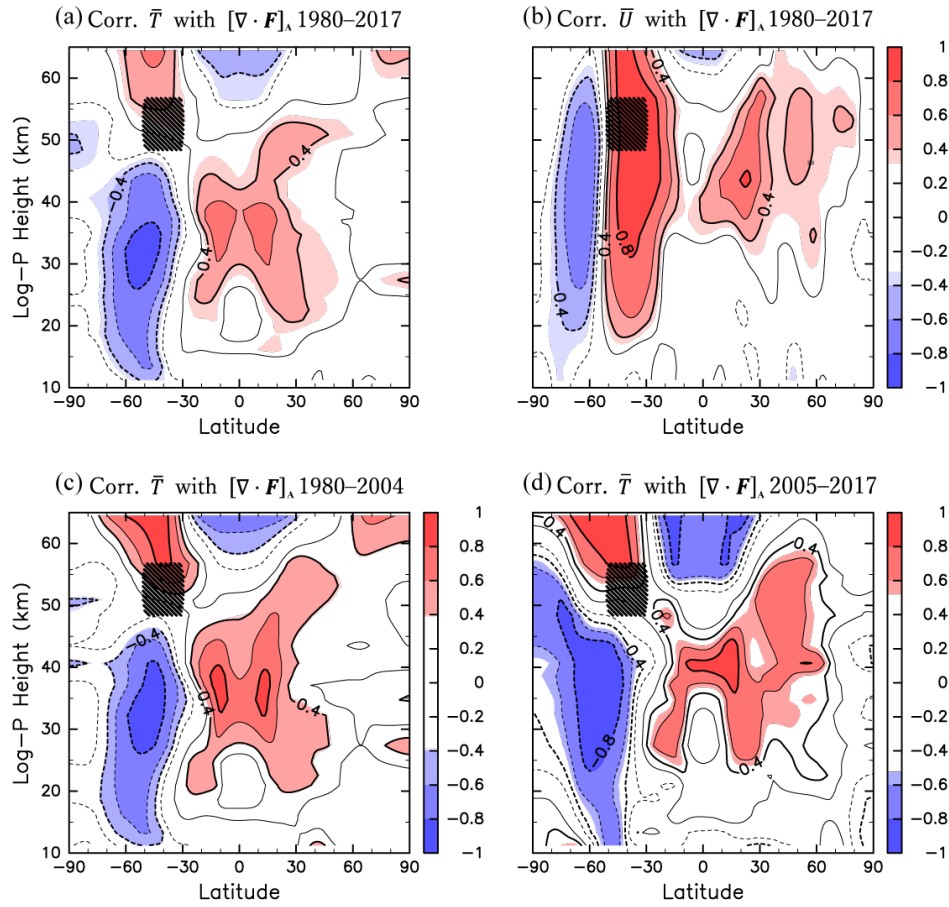

**Figure 3: Same as Fig. 2 but for the correlations of $[\nabla \cdot \mathbf{F}]_A$ with (a) $\overline{T}$ (1980–2017), (b) $\overline{u}$ (1980–2017), (c) $\overline{T}$ (1980–2004), and (d) $\overline{T}$ (2005–2017).**

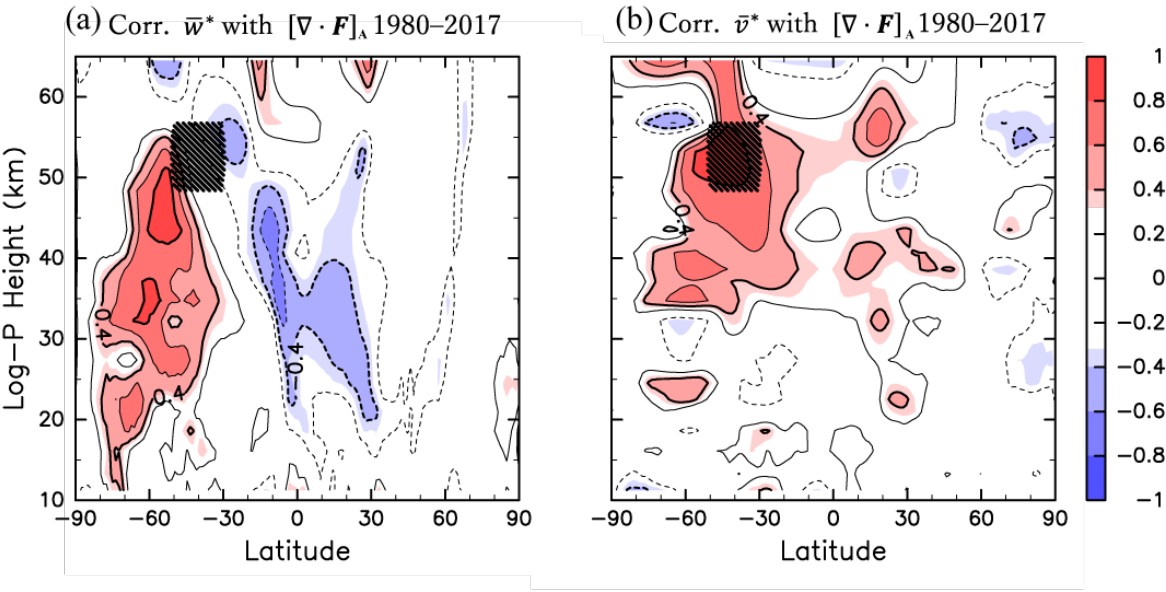


**Figure 4: Same as Fig. 2 but for the correlations of $[\nabla \cdot F]_A$ with (a) $\overline{w}^*$ (1980–2017) and $\overline{v}^*$ (1980–2017).**

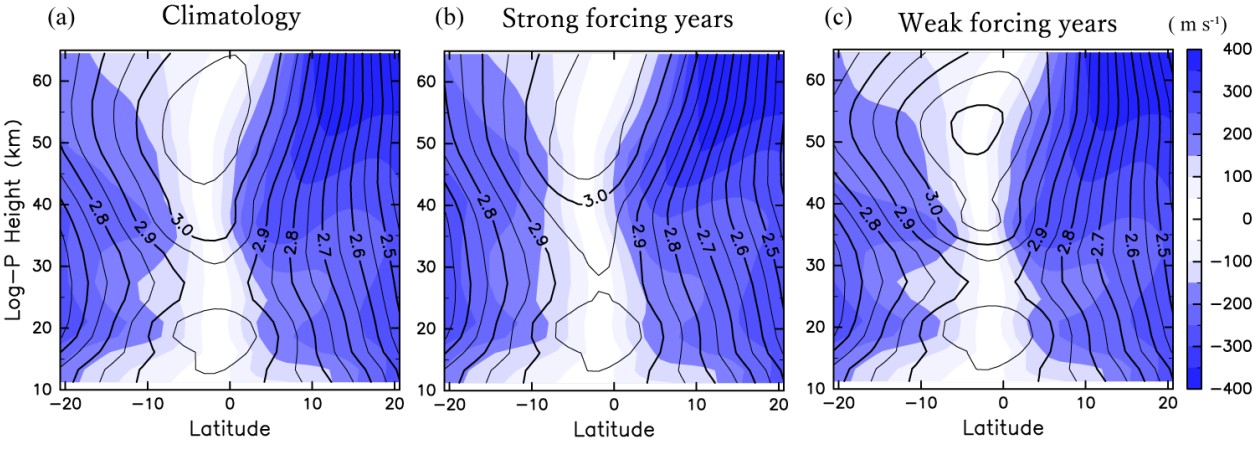

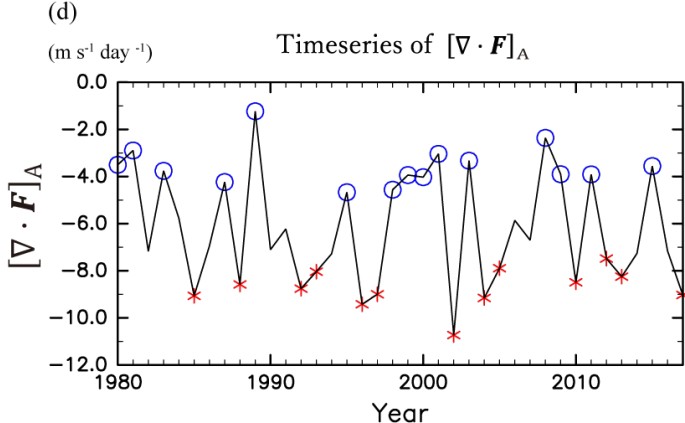

**Figure 5:** Meridional cross-sections of the (a) climatology of absolute angular momentum $\bar{M}$ (contours, $10^9$ m$^2$s$^{-1}$) and its meridional gradient $\bar{M}_y$ (colours, m s$^{-1}$) for 1980–2017, (b) composite for the strong wave forcing, and (c) composite for the weak wave forcing. Contour intervals are $5 \times 10^7$ m$^2$ s$^{-1}$. (d) Time series of $[\nabla \cdot F]_A$. The red and blue marks indicate the years used for (b) and (c), respectively.

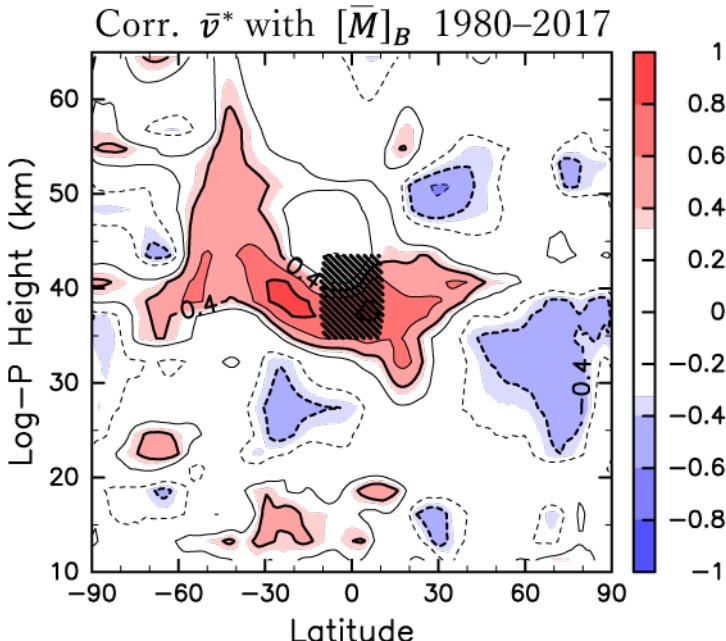


**Figure 6: Same as Fig.2 but for the correlation between $[M]_B$ and $\bar{v}^*$ (1980–2017). Region B is indicated by hatching.**

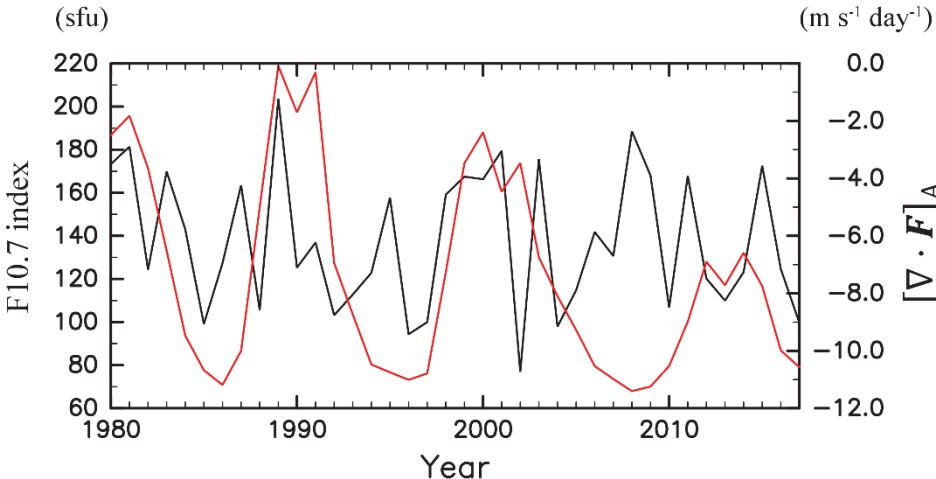

**Figure 7: Time series of the JJA mean F10.7 index (red, sfu $= 10^{-22}$ W m$^{-2}$ Hz$^{-1}$) and $[\nabla \cdot \mathbf{F}]_A$ (black, m s$^{-1}$ day$^{-1}$) for 1980–2017.**

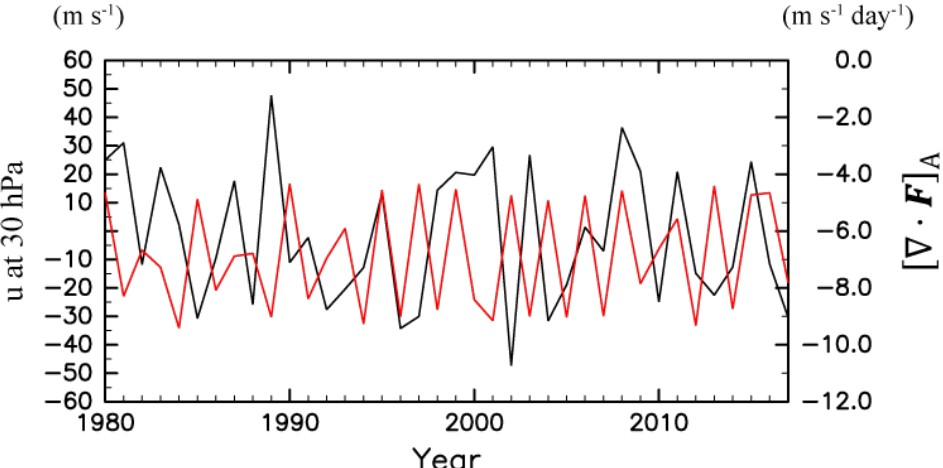

**Figure 8: Time series of the JJA mean $\bar{u}$ at the equator, 30 hPa (red, m s$^{-1}$) and $[\nabla \cdot \boldsymbol{F}]_A$ (black, m s$^{-1}$ day$^{-1}$) for 1980–2017.**

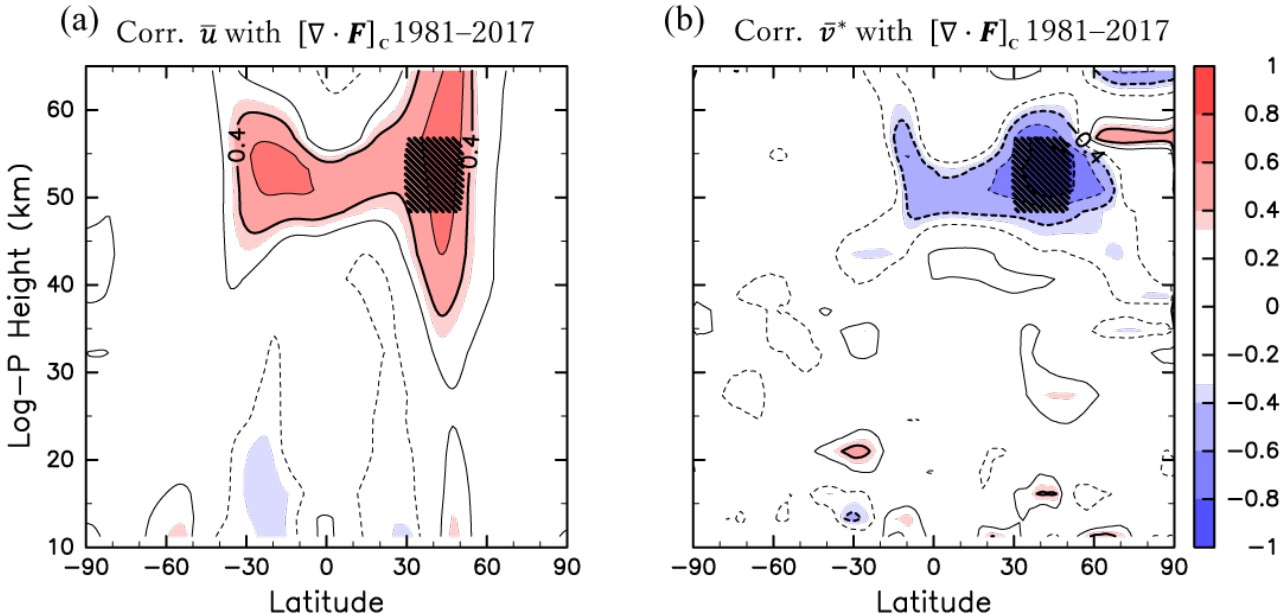

**Figure 9: Same as Fig. 2 but for the correlations of $[\nabla \cdot \boldsymbol{F}]_C$ with (a) $\overline{u}$ (1981–2017), (b) $\overline{v}^*$ (1981–2017) for DJF. Region C is indicated**
**by hatching.**