# Peer review of "Relation Between the Interannual Variability in the Stratospheric Rossby Wave Forcing and Zonal Mean Fields Suggesting an Interhemispheric Link in the Stratosphere"

_Annales Geophysicae, 2019_

## Referee Comment (RC1) · Anonymous Referee #1 · 20 Aug 2019

Review of "Relation Between the Interannual Variability in the Stratospheric Rossby Wave Forcing and Zonal Mean Fields Suggesting an Interhemispheric Link in the Stratosphere" by Matsushita et al.

General comments

This study uses MERRA-2 reanalysis to investigate the interannual variability in the stratosphere during the austral solstice season. The primary result is that there is a correlation of the 3-month average EP flux divergence in the winter midlatitude stratosphere with zonal mean fields in the tropics and summer hemisphere.

On the positive side, the paper shows that the seasonal mean anomalies of wave processes in the winter are correlated with anomalies in the zonal average dynamical fields in the tropics and low latitudes of both hemispheres. This type of pattern has been seen before in simple models of steady-state conditions and in a few analyses of observations, as cited in the manuscript. The new contribution of the present investigation focuses on the correlation patterns from a 38-year set of analyses. The authors are careful not to over-sell their results; since this is a study based on reanalysis, the ability to explore mechanisms is limited. However, they are able to link the signals they see to the expected results from a simple model.

The separation of the results into the period in which MERRA-2 includes or does not include assimilation of MLS is an important and useful component of the investigation.

While I do not have any major concerns about the analysis or about the explanations given by the authors, I wonder why the scope of the investigation was so narrow. Since all of the results use averages over the same 3-month portion of the year and only consider interannual variability of this a single time of year, this reader felt like the picture was incomplete. The comments below describe the questions I was left with after reading the manuscript.

Major comments

1. Why were 3-month averages used in the analysis? If shorter periods, such as individual months, were used, there would be a larger set of cases for investigating correlations. Did the authors probe the data to see whether correlation signals were stronger or weaker for averages over a subset of a season?

There is only a limited discussion of mechanism and I was unable to determine exactly what your interpretation is. The downward control mechanism that you refer to is valid for steady state (your three-month averages should be sufficient to satisfy this) but

does not give a circulation that extends very far away from the region of the forcing (see any of the steady state figures in Haynes et al 1991). It is plausible that a dynamical mechanism would have a timescale shorter than the season-spanning three-month period and still affect the season-average circulation. It seems that you do not support this interpretation since you say (l. 170-171) that the pattern you see is not a result of the same processes that cause SSW.

Since your analysis is limited to a single season, separated by nine months that are ignored, timescales of up to one year would be consistent with the results. Is external forcing responsible? You show that solar cycle forcing, which had been proposed in earlier studies, is not consistent with their results. Another "external" variation with a long timescale is the QBO; this can have an impact on circulation in the low latitude stratosphere. Looking at periods covering subsets of the three-month average would give some information about whether the signal has a timescale shorter than a year.

2. Related to the above, the definition of winter as June, July, August appears arbitrary. Wave forcing in the southern winter is spread over a long period from May to November or December. Why were these months chosen?

3. The statement (l. 26-27) that "the Rossby wave forcing in the winter extratropics cannot directly drive the cross-equatorial flow" needs more explanation since the results suggest to me that the Rossby wave forcing is driving the flow. Do you mean that this can only happen in certain circumstances that depend on the presence of an angular momentum gradient? Since there will generally be a region in the tropics where the gradient of M disappears, you seem to be implying that this wave driven circulation is not possible. Also, the statement later on (l. 142) seems to be saying something quite different: that the Rossby wave forcing is necessary to drive a circulation if there is an angular momentum gradient but a flow can exist without wave forcing if there is no angular momentum gradient. Please clarify.

Likewise, I found support lacking for your conclusion that "The cross-equatorial residual

mean flow is not directly driven by the Rossby wave forcing but indirectly maintained by the weak and small meridional gradient of the absolute angular momentum around the equator." Isn't it possible that the wave activity in the winter hemisphere, and the circulation response to it, is affecting M? Could you find cases where the wave forcing is high but M is large and vice versa? Otherwise, the relative impacts of these two processes cannot be separated.

4. It would be interesting to compare the opposite time of the year (northern winter) to determine whether a similar correlation exists then? Finding such a correlation would provide support that there is a physical mechanism rather than a chance correlation.

Minor comments

1. Some more description of the analysis is needed in Section 2. It took me a while to figure out that, when you discuss standard deviation, you mean only the standard deviation of fields that have already been averaged for the three months. The analysis for the wave amplitude is not clear – did you average daily amplitudes or daily Z'?

2. (l. 104) This sentence is not clear; Figure 1b does not show wave forcing. In fact, I wasn't sure what you are referring to in this entire paragraph. As far as I can tell, you have used the term "wave forcing" to mean something very specific (EP flux divergence in Area A) but it does not fit with the usage here.

3. (l. 113 ff) "This indicates that the wave forcing in the SH affects the mean fields in the low latitude region of the NH". Be careful here. Correlation does not mean causation. You need more evidence to say that one timeseries is affecting the other, rather than vice versa or both responding to some other forcing.

---

## Referee Comment (RC2) · Anonymous Referee #2 · 29 Aug 2019

This work examines the interhemispheric coupling in the stratosphere by conducting a statistical analysis of the interannual variability of Rossby wave forcing and mean temperature and zonal wind fields based on MERRA-2 reanalysis. The analysis suggests that the summer/northern stratosphere from the equator to the extratropical region is significantly correlated with the Rossby wave forcing in the winter/southern stratosphere. The authors also show that the meridional gradient of absolute angular momentum is generally near 0 around the equator, conducive of cross-equatorial Brewer-Dobson circulation and thus interhemispheric coupling. By comparing the interannual variability of wave forcing with solar variability, they show that the latter cannot completely explain the former.

The analysis is quite straightforward and the presentation is generally clear. It addresses an issue largely overlooked in previous studies. I have the following comments/questions for the authors:

1. The analysis focuses on JJA time period. How about boreal winter (DJF)? The same proposed mechanism should apply to the boreal winter too. I don't see any discussion of the stratospheric interhemispheric coupling in the text.

2. Figures 5(a-c) show that the width of the small meridiona gradient of absolute angular momentum around the equator varies along with wave forcing. It is not clear how significant this width variation–between 35 and 40 km the variation on each side is about 3 degrees latitude–for interhemispheric coupling. Is there any quantitative justification that this is (or is not) significant for the coupling? And related to my question 1, are the absolute angular momentum and its meridional gradient similar during boreal winter?

3. Figure 6 shows that the wave forcing is not always correlated with the solar activity. On the other hand, it seems over some time periods the forcing has a period of 2-3 years. And it is conceivable that the equatorial dynamical state (including the angular momentum/gradient) could be affected by QBO (probably comparable to solar impact, if not stronger). I wonder if this impact has been looked at in the analysis.

4. Page 7 line 197: "deflected from the midlatitudes". Please clarify whether the waves are deflected toward higher or lower latitudes.

---

## Author Comment (AC1) · 21 Oct 2019

**Response to the comments from Reviewer #1**

We greatly appreciate the reviewer for his/her critical reading and constructive comments. We have revised our manuscript as much as possible following his/her comments. Our response to each comment is described as follows:

**Response to major comments:**

[Figure]

1) *Why were 3-month averages used in the analysis? If shorter periods, such as individual months, were used, there would be a larger set of cases for investigating correlations. Did the authors probe the data to see whether correlation signals were stronger or weaker for averages over a subset of a season?*

We have added an explanation on the choice of the averaged period in Sect. 2, as follows:

"We analysed JJA-averaged fields to reduce the transient feature. We confirmed that the result for each month is qualitatively similar to those for JJA-averaged fields."

Significant correlation in the summer hemisphere is also observed for individual months although the correlation coefficient is small compared to that of JJA-averaged fields.

*There is only a limited discussion of mechanism and I was unable to determine exactly what your interpretation is. The downward control mechanism that you refer to is valid for steady state (your three-month averages should be sufficient to satisfy this) but does not give a circulation that extends very far away from the region of the forcing (see any of the steady state figures in Haynes et al 1991). It is plausible that a dynamical mechanism would have a timescale shorter than the season-spanning three-month period and still affect the season-average circulation. It seems that you do not support this interpretation since you say (l. 170-171) that the pattern you see is not a result of the same processes that cause SSW.*

We have revised Sect. 1 and Sect. 3. In Figure 2b, the interannual variability of the wave forcing averaged over Region A ($[\nabla \cdot F]_A$) is significantly correlated with the Rossby wave forcing around a 40 km altitude in the latitude range of 15°–70°S. It is indicated that the wave forcing in the subtropical region around 40 km shows similar interannual variability to that of $[\nabla \cdot F]_A$. Thus, the significant correlation of $\bar{v}^*$ observed in the region from the extratropics to subtropics of the SH can

be explained by the local balance between $-f\bar{v}^*$ (Coriolis force) and the Rossby wave forcing. Around the equator, on the other hand, meridional circulation can be maintained without wave forcing. An explanation has been added in the 1st paragraph of Section 1:

"However, the Rossby wave forcing in the winter extratropics does not directly drive the cross-equatorial flow around the equator since the wave forcing cannot be balanced with Coriolis force associated with meridional wind owing to small $f$. Although the meridional circulation in the extratropics requires wave forcing to cross angular momentum $(\bar{M})$ contours aligned nearly vertically, the meridional circulation can exist around the equator without wave forcing because the $\bar{M}$ contours are horizontally aligned."

*Since your analysis is limited to a single season, separated by nine months that are ignored, timescales of up to one year would be consistent with the results. Is external forcing responsible? You show that solar cycle forcing, which had been proposed in earlier studies, is not consistent with their results. Another "external" variation with a long timescale is the QBO; this can have an impact on circulation in the low latitude stratosphere. Looking at periods covering subsets of the three-month average would give some information about whether the signal has a timescale shorter than a year.*

Following the reviewer's suggestion, we have newly performed the analyses focusing on the relation of our results with the QBO. We used $\bar{u}$ at the equator and 30 hPa as a proxy of the QBO phase and made a new plot showing the correlation of the QBO phase and the wave forcing (Figure 8 in the revised manuscript). The correlation between $[\nabla \cdot F]_A$ and $\bar{u}$ at the equator and at 30 hPa is small and is not significant (-0.14). Since the reason why the correlation between the QBO and the wave forcing in Region A is insignificant is out of the scope of this study, we only note here that the height region for the wave forcing in the present study (namely, Region A) is located at a much higher altitude than that were focused in the previous studies (e.g., Baldwin and Dunkerton, 1998; Salby et al., 2011). We have
added a figure (Figure 8) and a paragraph on this point to Sect. 4.2.

2) *Related to the above, the definition of winter as June, July, August appears arbitrary. Wave forcing in the southern winter is spread over a long period from May to November or December. Why were these months chosen?*

We have added an explanation on the choice of the analysed period in Sect. 2, as mentioned before. We have also added a paragraph in Sect. 4.3 on the results for the extended period from May to November. Although the results for the extended time period in the winter SH are largely similar to those for JJA, the correlation coefficients become weak, especially in the summer NH, and the latitudinal extents of the statistically significant response of $\bar{u}$, $\bar{T}$, and $\bar{v}^*$ are limited up to the equator.

3) *The statement (l. 26-27) that "the Rossby wave forcing in the winter extratropics cannot directly drive the cross-equatorial flow" needs more explanation since the results suggest to me that the Rossby wave forcing is driving the flow. Do you mean that this can only happen in certain circumstances that depend on the presence of an angular momentum gradient? Since there will generally be a region in the tropics where the gradient of M disappears, you seem to be implying that this wave driven circulation is not possible. Also, the statement later on (l. 142) seems to be saying something quite different: that the Rossby wave forcing is necessary to drive a circulation if there is an angular momentum gradient but a flow can exist without wave forcing if there is no angular momentum gradient. Please clarify.*

We use the phrase "directly drive" in the sense that the Rossby wave forcing is balanced with the Coriolis force for meridional wind to maintain $\bar{v}^*$. The phrase "cross-equatorial flow" means the meridional flow at low latitudes which crosses the equator, and does not mean the whole circulation from the equator to polar latitudes. We have revised the sentence (ll. 26–27 in the original manuscript) to clarify the meaning, as follows:

"the Rossby wave forcing in the winter extratropics does not directly drive the crossequatorial flow around the equator since the wave forcing cannot be balanced with the Coriolis force for the meridional wind."

*Likewise, I found support lacking for your conclusion that "The cross-equatorial residual mean flow is not directly driven by the Rossby wave forcing but indirectly maintained by the weak and small meridional gradient of the absolute angular momentum around the equator." Isn't it possible that the wave activity in the winter hemisphere, and the circulation response to it, is affecting M? Could you find cases where the wave forcing is high but M is large and vice versa? Otherwise, the relative impacts of these two processes cannot be separated.*

We have cited Semeniuk and Shepherd (2002) and added a paragraph to Sect. 3. They examined the middle-atmosphere Hadley circulation and its interaction with extratropical wave-driven circulation, using a numerical model. They showed that the extratropical wave-driven circulation affects the meridional gradient of angular momentum ($\bar{M}_y$) around the equator together with the middle-atmosphere Hadley circulation, and that the significant overturning of $\bar{M}$ contours around the equator is attributable to the combination of the middle-atmosphere Hadley circulation and the extratropical wave-driven circulation. We have also added the results of the analyses on the $\bar{M}_y$ around the equator to Sect. 3. The correlation of $[\nabla \cdot F]_A$ with the $\bar{M}$ averaged over the region of the cross-equatorial flow (10°S–10°N, 35–45 km) is significantly positive (0.49). The wave forcing in the Region A is likely to drive the residual mean flow in the extratropics and subtropics of the SH and to modify the mean zonal wind around the equator with a small $\left|\bar{M}_y\right|$, and the present study does not intend to separate these processes.

4) *It would be interesting to compare the opposite time of the year (northern winter) to determine whether a similar correlation exists then? Finding such a correlation would provide support that there is a physical mechanism rather than a chance correlation.*

We have added a paragraph in Sect. 4.3 and figures as Fig. 9 in the revised
manuscript regarding the results for the northern winter. It is indicated that the interhemispheric link and cross-equatorial flow in the boreal winter is associated with the wave forcing in the NH stratosphere as well, while the latitudinal extent to the summer hemisphere is limited compared to that in the austral winter. Due to significantly large amplitude of planetary waves in the NH winter, which sometimes causes the breakdown of the polar vortex, a linear relation is unlikely obtained between the wave forcing and mean fields in the NH winter.

**Response to minor comments**:

1) *Some more description of the analysis is needed in Section 2. It took me a while to figure out that, when you discuss standard deviation, you mean only the standard deviation of fields that have already been averaged for the three months. The analysis for the wave amplitude is not clear – did you average daily amplitudes or daily Z'?*

> We have revised the sentence (l. 82 in the original manuscript) as follows: "Figure 1c shows the climatology of three-hourly values of the root mean square of the geopotential height deviation ($Z\prime$) from the zonal mean obtained at every three hours "

2) *This sentence is not clear; Figure 1b does not show wave forcing. In fact, I wasn't sure what you are referring to in this entire paragraph. As far as I can tell, you have used the term "wave forcing" to mean something very specific (EP flux divergence in Area A) but it does not fit with the usage here.*

> We have corrected the number of the figure in the sentence (l.104 in the original manuscript) from 1b to 1d.

3) *"This indicates that the wave forcing in the SH affects the mean fields in the low latitude region of the NH". Be careful here. Correlation does not mean causation. You need more evidence to say that one timeseries is affecting the other, rather than vice versa or both responding to some other forcing.*

We have revised the sentence as followings:

"This indicates that the wave forcing in the SH is related to the mean fields in the low latitude region of the NH"

References:

Baldwin, M. P. and Dunkerton, T. J.: Quasi-biennial modulation of the southern hemisphere stratospheric polar vortex, Geophys. Res. Lett., 25(17), 3343–3346, doi:10.1029/98GL02445, 1998.

Semeniuk, K. and Shepherd, T. G.: The Middle-Atmosphere Hadley Circulation and Equatorial Inertial Adjustment, J. Atmos. Sci., 58(21), 3077–3096, doi:10.1175/1520-0469(2001)058<3077:TMAHCA>2.0.CO;2, 2001.

Salby, M., Titova, E. and Deschamps, L.: Rebound of Antarctic ozone, Geophys. Res. Lett., 38(9), doi:10.1029/2011GL047266, 2011.

---

## Author Comment (AC2) · 21 Oct 2019

**Response to the comments from Reviewer #2**

We greatly appreciate the reviewer for his/her critical reading and constructive comments. We have revised our manuscript as much as possible following his/her comments. Our response to each comment is described as follows:

1)*The analysis focuses on JJA time period. How about boreal winter (DJF)? The same proposed mechanism should apply to the boreal winter too. I don't see any discussion*

[Figure]

*of the stratospheric interhemispheric coupling in the text.*

Following the reviewer's suggestion, we have newly performed the analyses for the boreal winter (DJF). As a result, it is indicated that the interhemispheric link and cross-equatorial flow in the boreal winter is associated with the wave forcing in the NH stratosphere as well, while the latitudinal extent to the summer hemisphere is limited compared to that in the austral winter. Due to significantly large amplitude of planetary waves in the NH winter, which sometimes causes the breakdown of the polar vortex, a linear relation is unlikely obtained between the wave forcing and mean fields in the NH winter. We have newly added Sect.4.3 on the results for the boreal winter and have added figures as Fig. 9 in the revised manuscript.

2) *Figures 5(a-c) show that the width of the small meridional gradient of absolute an-gular momentum around the equator varies along with wave forcing. It is not clear how significant this width variation–between 35 and 40 km the variation on each side is about 3 degrees latitude–for interhemispheric coupling. Is there any quantitative justification that this is (or is not) significant for the coupling?*

In order to clarify the relation between meridional gradient of absolute angular momentum ($\bar{M}_y$) around the equator and cross-equatorial flow, we define the region of 10°S–10°N, 35–45 km as Region B, and examine $\bar{M}$ averaged over Region B (hereafter referred to as $\left[\bar{M}\right]_B$). The correlation between $\left[\nabla \cdot F\right]_A$ and $\left[\bar{M}\right]_B$ is significantly positive (0.49), which is consistent with the results of the composite analyses (Figure 5 in the revised manuscript). The correlation between the inter-annual variability of $\left[\bar{M}\right]_B$ and $\bar{v}^*$ is significantly highly positive in the region of the cross-equatorial flow as seen in Fig. 4b. Thus, when the absolute angular mo-mentum at the Region B is small, the southward cross-equatorial flow through the Region B is strong. We have added a paragraph and a figure (Fig. 6 in the revised manuscript) on the $\bar{M}_y$ around the equator to Sect.3.

*And related to my question 1, are the absolute angular momentum and its meridional*

*gradient similar during boreal winter?*

The correlation of $\bar{u}$ and extratropical stratospheric wave forcing in DJF, shown in Fig. 9a, is significantly positive and higher than 0.4 at 50–55 km and 30°S–30°N, and the correlation of $\bar{v}^*$ is also significant at 50–55 km and 20°S–30°N. At the equator, $\bar{u}$, and thus absolute angular momentum, is small when the Rossby wave forcing in the boreal winter stratosphere is strong in DJF as in JJA, although the altitudes where these variabilities are observed is different from that of JJA.

3) *Figure 6 shows that the wave forcing is not always correlated with the solar activity. On the other hand, it seems over some time periods the forcing has a period of 2-3 years. And it is conceivable that the equatorial dynamical state (including the angular momentum/gradient) could be affected by QBO (probably comparable to solar impact, if not stronger). I wonder if this impact has been looked at in the analysis.*

Following the reviewer's suggestion, we have newly performed the analyses focusing on the relation of our results with the QBO. We used $\bar{u}$ at the equator and 30 hPa as a proxy of the QBO phase and made a new plot showing the correlation of the QBO phase and the wave forcing (Figure 8 in the revised manuscript). The correlation between $[\nabla \cdot F]_A$ and $\bar{u}$ at the equator and at 30 hPa is small and is not significant (-0.14). Since the reason why the correlation between the QBO and the wave forcing in Region A is insignificant is out of the scope of this study, we only note here that the height region for the wave forcing in the present study (namely, Region A) is located at a much higher altitude than that were focused in the previous studies (e.g., Baldwin and Dunkerton, 1998; Salby et al., 2011). We have added a figure (Figure 8) and a paragraph on this point to Sect. 4.2.

4) *Page 7 line 197: "deflected from the midlatitudes". Please clarify whether the waves are deflected toward higher or lower latitudes.*

We have revised the phrase as follows: "deflected from the midlatitudes to higher latitudes"

[Figure]

**References:**

Baldwin, M. P. and Dunkerton, T. J.: Quasi-biennial modulation of the southern hemisphere stratospheric polar vortex, Geophys. Res. Lett., 25(17), 3343–3346, doi:10.1029/98GL02445, 1998.

Salby, M., Titova, E. and Deschamps, L.: Rebound of Antarctic ozone, Geophys. Res. Lett., 38(9), doi:10.1029/2011GL047266, 2011.
* * *

---

## Author Response (AR1)

Response to the comments from Reviewer #1

We greatly appreciate the reviewer for his/her thorough review and constructive comments. We have revised our manuscript as much as possible following his/her comments. Our response to each comment is described as follows:

**Response to major comments:**

1) *Why were 3-month averages used in the analysis? If shorter periods, such as individual months, were used, there would be a larger set of cases for investigating correlations. Did the authors probe the data to see whether correlation signals were stronger or weaker for averages over a subset of a season?*

We have added explanations on the choice of the averaged period in Sect. 2, as follows:
"We analysed JJA-averaged fields to reduce the transient feature, and confirmed that the results for the respective month are qualitatively similar to those for JJA-averaged fields."
The correlation in the summer hemisphere is also observed for individual months although the correlation coefficient is small compared to that of JJA-averaged fields.

*There is only a limited discussion of mechanism and I was unable to determine exactly what your interpretation is. The downward control mechanism that you refer to is valid for steady state (your three-month averages should be sufficient to satisfy this) but does not give a circulation that extends very far away from the region of the forcing (see any of the steady state figures in Haynes et al 1991). It is plausible that a dynamical mechanism would have a timescale shorter than the season-spanning three-month period and still affect the season-average circulation. It seems that you do not support this interpretation since you say (l. 170-171) that the pattern you see is not a result of the same processes that cause SSW.*

We have revised Sect. 1 and Sect. 3. In Figure 2b, the interannual variability of the wave forcing averaged over Region A ($[\nabla \cdot \boldsymbol{F}]_A$) is significantly correlated with the Rossby wave forcing around 40 km altitude in the latitude range of 70°–15°S. It is indicated that the wave forcing in the subtropical region around 40 km shows similar interannual variability to that of $[\nabla \cdot \boldsymbol{F}]_A$. Thus, the significant correlation of $\bar{v}^*$ from the extratropics to subtropics of the SH can be explained by the local balance between $-f\bar{v}^*$ (Coriolis force) and the Rossby wave forcing. Around the equator, on the other hand, meridional circulation can be maintained without wave forcing as added explanation in the 1st paragraph of Section 1, as follows:

"However, the Rossby wave forcing in the winter extratropics does not directly drive the cross-equatorial flow around the equator since the wave forcing cannot be balanced with Coriolis force associated with meridional wind. While the meridional circulation in the extratropics requires wave forcing to cross angular momentum ($\overline{\mathrm{M}}$) contours, which are aligned nearly vertically, around the equator, the meridional circulation can exist without wave forcing due to the horizontally aligned $\overline{\mathrm{M}}$ contours as far as the conservation of mass is satisfied. "

*Since your analysis is limited to a single season, separated by nine months that are ignored, timescales of up to one year would be consistent with the results. Is external forcing responsible? You show that solar cycle forcing, which had been proposed in earlier studies, is not consistent with their results. Another "external" variation with a long timescale is the QBO; this can have an impact on circulation in the low latitude stratosphere. Looking at periods covering subsets of the three-month average would give some information about whether the signal has a timescale shorter than a year.*

Following the reviewer's suggestion, we have newly performed the analyses focusing on the relation of our results with the QBO. We use $\bar{u}$ at the equator and 30 hPa as a proxy of the QBO phase and made a new plot showing the correlation of the QBO phase and the wave forcing (Figure 8 in the revised manuscript). The correlation between $[\nabla \cdot \boldsymbol{F}]_A$ and $\bar{u}$ at the equator and at 30 hPa is small and is not significant (-0.14). Since the reason why the correlation between the QBO and the wave forcing in Region A is insignificant is out of the scope of this study, we only note here that the height region for the wave forcing in the present study (namely, Region A) is located at a much higher altitude than that were focused in the previous studies (e.g., Baldwin and Dunkerton, 1998; Salby et al., 2011). We have added a figure (Figure 8) and a paragraph on that to Sect. 4.2.

2) *Related to the above, the definition of winter as June, July, August appears arbitrary. Wave forcing in the southern winter is spread over a long period from May to November or December. Why were these months chosen?*

We have added an explanation on the choice of the analysed period in Sect. 2, as mentioned before. We have also added a paragraph in Sect. 4.3 on the results for the extended period from May to November. Although the results for the extended time period in the winter SH are largely similar to those for JJA, the correlation coefficients become weak, especially in the summer NH, and the latitudinal extents of the statistically significant response of $\bar{u}$, $\bar{T}$, and $\bar{v}^*$ are limited up to the equator.

3) *The statement (l. 26-27) that "the Rossby wave forcing in the winter extratropics cannot directly drive the cross-equatorial flow" needs more explanation since the results suggest to me that the Rossby wave forcing is driving the flow. Do you mean that this can only happen in certain circumstances that depend on the presence of an angular momentum gradient? Since there will generally be a region in the tropics where the gradient of M disappears, you seem to be implying that this wave driven circulation is not possible. Also, the statement later on (l. 142) seems to be saying something quite different: that the Rossby wave forcing is necessary to drive a circulation if there is an angular momentum gradient but a flow can exist without wave forcing if there is no angular momentum gradient. Please clarify.*

We use the phrase "directly drive" in the sense that the Rossby wave forcing is balanced with the Coriolis force with meridional wind to maintain $\bar{v}^*$. The phrase "cross-equatorial flow" means the meridional flow at low latitudes which crosses the equator, and does not mean the whole circulation from equator to polar latitudes. We have revised the sentence (l. 26–27 in the original manuscript) to clarify the meaning, as follows:

"the Rossby wave forcing in the winter extratropics does not directly drive the cross-equatorial flow around the equator since the wave forcing cannot be balanced with Coriolis force associated with meridional wind."

*Likewise, I found support lacking for your conclusion that "The cross-equatorial residual mean flow is not directly driven by the Rossby wave forcing but indirectly maintained by the weak and small meridional gradient of the absolute angular momentum around the equator." Isn't it possible that the wave activity in the winter hemisphere, and the circulation response to it, is affecting M? Could you find cases where the wave forcing is high but M is large and vice versa? Otherwise, the relative impacts of these two processes cannot be separated.*

We have added Semeniuk and Shepherd (2002) to the reference and added a paragraph on that to Sect. 3. They examined the middle-atmosphere Hadley circulation and its interaction with extratropical wave-driven circulation, using a numerical model. They showed that the extratropical wave-driven circulation affects the meridional gradient of angular momentum $(\bar{M}_y)$ around the equator together with the middle-atmosphere Hadley circulation, and that the significant overturning of $\bar{M}$ contours around the equator is attributable to the combination of the middle-atmosphere Hadley circulation and the extratropical wave-driven circulation. We have also added the results of the analyses on the $\bar{M}_y$ around the equator to Sect. 3. The correlation of $[\nabla \cdot \boldsymbol{F}]_A$ with the $\bar{M}$ averaged over the region where the cross-equatorial flow (10°S–10°N, 35–45 km) is significantly positive (0.49).

The wave forcing in the Region A is likely to drive the residual mean flow in the extratropics and subtropics of the SH and to modify the mean wind around the equator with a small $\left|\overline{M}_y\right|$, and the present study does not intend to separate these processes.

4) *It would be interesting to compare the opposite time of the year (northern winter) to determine whether a similar correlation exists then? Finding such a correlation would provide support that there is a physical mechanism rather than a chance correlation.*

We have added a paragraph in Sect. 4.3 and figures as Fig. 9 in the revised manuscript on the results for the northern winter. It is indicated that the interhemispheric link and cross-equatorial flow in the boreal winter is associated with the wave forcing in the NH stratosphere while the latitudinal extent to the summer hemisphere is limited compared to the austral winter. Due to the large amplitude of planetary wave in the NH winter, which sometimes causes the breakdown of the polar vortex, a linear relation is unlikely to be obtained between the wave forcing and mean fields in the NH winter.

**Response to minor comments:**

1) *Some more description of the analysis is needed in Section 2. It took me a while to figure out that, when you discuss standard deviation, you mean only the standard deviation of fields that have already been averaged for the three months. The analysis for the wave amplitude is not clear – did you average daily amplitudes or daily Z'?*

We have revised the sentence (l. 82 in the original manuscript) as follows:
"Figure 1c shows the climatology of three-hourly values of the root mean square of the geopotential height deviation ($Z'$) from the zonal mean"

2) *This sentence is not clear; Figure 1b does not show wave forcing. In fact, I wasn't sure what you are referring to in this entire paragraph. As far as I can tell, you have used the term "wave forcing" to mean something very specific (EP flux divergence in Area A) but it does not fit with the usage here.*

We have corrected the number of the figure in the sentence (l.104 in the original manuscript) from 1b to 1d.

3) *"This indicates that the wave forcing in the SH affects the mean fields in the low latitude region of the NH". Be careful here. Correlation does not mean causation. You need more evidence to say that one timeseries is affecting the other, rather than vice versa or both responding to some other forcing.*

We have revised the sentence as followings:

"This indicates that the wave forcing in the SH is related to the mean fields in the low latitude region of the NH"

In order to clarify the relation between meridional gradient of absolute angular momentum ($\bar{M}_y$) around the equator and cross-equatorial flow, we define the region of 10°S–10°N, 35–45 km as Region B, and examine $\bar{M}$ averaged over Region B (hereafter referred to as $[\bar{M}]_B$). The correlation between $[\nabla \cdot \boldsymbol{F}]_A$ and $[\bar{M}]_B$ is significantly positive (0.49), which is consistent with the results of the composite analyses (Figure 5 in the revised manuscript). The correlation between the interannual variability of $[\bar{M}]_B$ and $\bar{v}^*$ is high and significantly positive in the region of the cross-equatorial flow indicated in Fig. 4b. Thus, when the absolute angular momentum at the Region B is small, the southward cross-equatorial flow through the Region B is strong. We have added a paragraph and a figure (Fig. 6 in the revised manuscript) on the $\bar{M}_y$ around the equator to Sect.3.

*And related to my question 1, are the absolute angular momentum and its meridional gradient similar*

*during boreal winter?*

The correlation of $\bar{u}$ and extratropical stratospheric wave forcing in DJF, shown in Fig. 9a, is significantly positive and larger than 0.4 at ~50–55 km and ~30°S–30°N, and the correlation of $\bar{v}^*$ is also significant at ~50–55 km and ~20°S–30°N. At the equator, $\bar{u}$, and thus absolute angular momentum, is small when the Rossby wave forcing at the boreal winter stratosphere is strong in DJF as in JJA, although the altitudes of these variability is different from that of JJA.

3) *Figure 6 shows that the wave forcing is not always correlated with the solar activity. On the other hand, it seems over some time periods the forcing has a period of 2-3 years. And it is conceivable that the equatorial dynamical state (including the angular momentum/gradient) could be affected by QBO (probably comparable to solar impact, if not stronger). I wonder if this impact has been looked at in the analysis.*

Following the reviewer's suggestion, we have newly performed the analyses focusing on the relation of our results with the QBO. We use $\bar{u}$ at the equator and 30 hPa as a proxy of the QBO phase and made a new plot showing the correlation of the QBO phase and the wave forcing (Figure 8 in the revised manuscript). The correlation between $[\nabla \cdot F]_A$ and $\bar{u}$ at the equator and at 30 hPa is small and is not significant (-0.14). Since the reason why the correlation between the QBO and the wave forcing in Region A is insignificant is out of the scope of this study, we only note here that the height region for the wave forcing in the present study (namely, Region A) is located at a much higher altitude than that were focused in the previous studies (e.g., Baldwin and Dunkerton, 1998; Salby et al., 2011). We have added a figure (Figure 8) and a paragraph on that to Sect. 4.2.

4) *Page 7 line 197: "deflected from the midlatitudes". Please clarify whether the waves are deflected toward higher or lower latitudes.*

We have revised the phrase as follows:
"deflected from the midlatitudes to higher latitudes"

References:

[revised manuscript text omitted]
 $\overline{\overline{M}}$ (contours, $10^9\ \mathrm{m^2 s^{-1}}$) and its meridional gradient $\overline{M}_y$ (colours, $\mathrm{m\ s^{-1}}$) for 1980–2017, (b) composite for the strong wave forcing, and (c) composite for the weak wave forcing. Contour intervals are $5 \times 10^7\ \mathrm{m^2\ s^{-1}}$. (d) Time series of $[\nabla \cdot \boldsymbol{F}]_A$. The red and blue marks indicate the years used for (b) and (c), respectively.

[Figure]

405

Figure 6

[Figure]

**Figure 6: Same as Fig.2 but for the correlation between $[\overline{M}]_B$ and $\overline{v}^*$ (1980–2017). Region B is indicated by hatching.**

[revised manuscript text omitted]

modifies

| Page 5: Inserted | Yuki Matsushita | 11/8/2019 11:21:00 AM |

in the Region A  is likely to modify

| Page 5: Inserted | Yuki Matsushita | 11/8/2019 11:21:00 AM |
| --- | --- | --- |

composite

| Page 5: Deleted | Yuki Matsushita | 11/8/2019 11:21:00 AM |
| --- | --- | --- |

**Relationship**

| Page 5: Inserted | Yuki Matsushita | 11/8/2019 11:21:00 AM |
| --- | --- | --- |

**Relation**

| Page 6: Deleted | Yuki Matsushita | 11/8/2019 11:21:00 AM |
| --- | --- | --- |

**11-year solar cycle**

| Page 6: Inserted | Yuki Matsushita | 11/8/2019 11:21:00 AM |
| --- | --- | --- |

**external forcing**

| Page 6: Deleted | Yuki Matsushita | 11/8/2019 11:21:00 AM |
| --- | --- | --- |

relationship

| Page 6: Inserted | Yuki Matsushita | 11/8/2019 11:21:00 AM |
| --- | --- | --- |

relation

| Page 6: Deleted | Yuki Matsushita | 11/8/2019 11:21:00 AM |
| --- | --- | --- |

.

| Page 6: Inserted | Yuki Matsushita | 11/8/2019 11:21:00 AM |
| --- | --- | --- |

to higher latitudes.

| Page 6: Deleted | Yuki Matsushita | 11/8/2019 11:21:00 AM |
| --- | --- | --- |

| Page 6: Inserted | Yuki Matsushita | 11/8/2019 11:21:00 AM |
| --- | --- | --- |

| Page 7: Inserted | Yuki Matsushita | 11/8/2019 11:21:00 AM |
| --- | --- | --- |

The QBO can also modulate the extratropical circulation and the Rossby wave in the winter hemisphere. Following Salby et al. (2011), we use $\bar{u}$ at the equator and 30 hPa as a proxy of the QBO phase. Figure 8 shows the time series of $[\nabla \cdot \boldsymbol{F}]_{\mathrm{A}}$ and of $\bar{u}$ at the equator, 30 hPa. Although both time series show short-term variability,

their typical periods of oscillation seem to be different, and the correlation is small and is not significant (-0.14). This result is consistent with that in Fig. 3b since the correlation between $[\nabla \cdot \boldsymbol{F}]_A$ and $\bar{u}$ is also small and not significant around the equator in the altitude range of 15-25 km. In the present analyses for the JJA-averaged fields, there is no significant relation between the Rossby wave forcing at Region A and the QBO on the interannual time scale. Since the reason why the correlation between the QBO and the wave forcing in Region A is insignificant is out of the scope of this study, we only note here that the height region for the wave forcing in the present study (namely, Region A) is located at a much higher altitude than that was focused in the previous studies (e.g., Baldwin and Dunkerton, 1998; Salby et al., 2011).

**4.3 Results for zonal mean fields averaged over May through November and boreal winter**

In the present study, we have examined the JJA-averaged field in terms of the interannual variability of the Rossby wave forcing and its relation to zonal mean fields. The same analysis is performed for the fields averaged over May through November, when the Rossby wave in the stratosphere are active in the SH. Although the results for the extended time period in the winter SH are quite similar to those for JJA, the correlation coefficients are weak, especially in the summer NH, and the statistically significant response of $\bar{u}$, $\bar{T}$, and $\bar{v}^*$ is limited only up to the equator. This is likely because the zonal mean fields and wave forcing in the NH in May and the SH in November are largely different (e.g., Randel, 1988). Furthermore, the tendency of the zonal mean fields is not negligible in the zonal momentum equation for the equinoctial seasons. Because the seasonal evolution of the zonal mean fields is responsible for that of radiative heating, the temporal change of the zonal mean fields in JJA is in general small compared with the equinoctial season (Sato and Hirano, 2018), which is a preferable condition for the downward control principle analyses. As a result, the correlation analyses for JJA clearly indicate the interhemispheric link between the wave forcing in the winter SH and zonal mean fields in the summer NH.

Last but not the least, results for the interhemispheric link in the boreal winter are briefly described. To examine the interhemispheric link in the NH winter season, zonal mean fields averaged over December to February (DJF) of 1981–2017 are analysed. From a latitude-pressure section of the climatological wave forcing in DJF (not shown), it is seen that the wave forcing has a maximum in the region of 0.3–1 hPa and 30°–50°N, hereafter referred to as Region C. The correlation of DJF-averaged $\bar{u}$ with the wave forcing averaged over Region C ($[\nabla \cdot \boldsymbol{F}]_C$) is shown in Fig. 9a. Although the correlation is significantly positive around Region C and in 30°S–30°N, 45–55 km, the correlation is not statistically significant at the latitudes higher than 30°S. Figure 9b shows the correlation of DJF-averaged $\bar{v}^*$ with $[\nabla \cdot \boldsymbol{F}]_C$. The correlation is significantly negative at Region C and at 10°S–30°N. It is indicated that the interhemispheric link and cross-equatorial flow in the boreal winter is associated with the wave forcing in Region C, while the latitudinal extent to the summer SH is limited compared to the austral winter (Figs. 3a and 4b). The difference in the correlation between JJA and DJF may be explained by the linearity of the response of the zonal mean fields to the wave forcing. Due to large amplitude

of planetary waves in the NH winter, which sometimes causes the breakdown of the polar vortex, a linear relation is unlikely obtained between the wave forcing and mean fields in the NH winter. The detailed analysis of the NH winter is beyond the scope of this paper.

| Page 7: Deleted | Yuki Matsushita | 11/8/2019 11:21:00 AM |

relationship

| Page 7: Inserted | Yuki Matsushita | 11/8/2019 11:21:00 AM |

relation

| Page 8: Deleted | Yuki Matsushita | 11/8/2019 11:21:00 AM |

that

| Page 8: Inserted | Yuki Matsushita | 11/8/2019 11:21:00 AM |

at the SH mid-latitudes, and this circulation

| Page 8: Deleted | Yuki Matsushita | 11/8/2019 11:21:00 AM |

weak

| Page 8: Inserted | Yuki Matsushita | 11/8/2019 11:21:00 AM |

mass-continuity

| Page 8: Inserted | Yuki Matsushita | 11/8/2019 11:21:00 AM |

| Page 9: Deleted | Yuki Matsushita | 11/8/2019 11:21:00 AM |

2001b

| Page 9: Inserted | Yuki Matsushita | 11/8/2019 11:21:00 AM |

2001

| Page 9: Inserted | Yuki Matsushita | 11/8/2019 11:21:00 AM |

Salby, M., Titova, E. and Deschamps, L.: Rebound of Antarctic ozone, Geophys. Res. Lett., 38(9), doi:10.1029/2011GL047266, 2011.

Sato, K. and Hirano, S.: The climatology of the Brewer–Dobson circulation and the contribution of gravity waves, Atmos. Chem. Phys., 19(7), 4517–4539, doi:10.5194/acp-19-4517-2019, 2019.

| Page 10: Inserted | Yuki Matsushita | 11/8/2019 11:21:00 AM |

Semeniuk, K. and Shepherd, T. G.: The Middle-Atmosphere Hadley Circulation and Equatorial Inertial Adjustment, J. Atmos. Sci., 58(21), 3077–3096, doi:10.1175/1520-0469(2001)058<3077:TMAHCA>2.0.CO;2, 2001.

| Page 11: Deleted | Yuki Matsushita | 11/8/2019 11:21:00 AM |

[Figure]

[Figure]

[Figure]

| Page 12: Inserted | Yuki Matsushita | 11/8/2019 11:21:00 AM |

[Figure]

| Page 12: Deleted | Yuki Matsushita | 11/8/2019 11:21:00 AM |

.

| Page 12: Inserted | Yuki Matsushita | 11/8/2019 11:21:00 AM |

, and the zero contours are omitted.

Page 13: Deleted        Yuki Matsushita        11/8/2019 11:21:00 AM

[Figure]

Page 13: Inserted        Yuki Matsushita        11/8/2019 11:21:00 AM

(a) Corr. $\bar{T}$ with $[\nabla \cdot \boldsymbol{F}]_{\mathrm{A}}$ 1980–2017

(b) Corr. $\bar{U}$ with $[\nabla \cdot \boldsymbol{F}]_{\mathrm{A}}$ 1980–2017

(c) Corr. $\bar{T}$ with $[\nabla \cdot \boldsymbol{F}]_{\mathrm{A}}$ 1980–2004

(d) Corr. $\bar{T}$ with $[\nabla \cdot \boldsymbol{F}]_{\mathrm{A}}$ 2005–2017

| Page 13: Deleted | Yuki Matsushita | 11/8/2019 11:21:00 AM |
|---|---|---|

in

| Page 14: Deleted | Yuki Matsushita | 11/8/2019 11:21:00 AM |
|---|---|---|

(a) Climatology  (b) Strong forcing years  (c) Weak forcing years  ( m s⁻¹)

[Figure]

(d)
(m s⁻¹ day⁻¹)

Timeseries of $[\nabla \cdot \boldsymbol{F}]_A$

[Figure]

Figure 6

Page 15: Inserted                    Yuki Matsushita                    11/8/2019 11:21:00 AM

[Figure]

**Figure 6: Same as Fig.2 but for the correlation between $[M]_B$ and $\bar{v}^*$ (1980–2017). Region B is indicated by hatching.**

[Figure]

**Figure 7**

Page 17: Inserted          Yuki Matsushita          11/8/2019 11:21:00 AM

[Figure]

**Figure 8: Time series of the JJA mean $\bar{u}$ at the equator, 30 hPa (red, m s$^{-1}$) and $[\nabla \cdot \boldsymbol{F}]_A$ (black, m s$^{-1}$ day$^{-1}$) for 1980–2017.**

[Figure]

**Figure 9: Same as Fig. 2 but for the correlations of $[\nabla \cdot F]_C$ with (a) $\overline{u}$ (1981–2017), (b) $\overline{v}^*$ (1981–2017) for DJF. Region C is indicated by hatching.**

Header and footer changes

Text Box changes

Header and footer text box changes

Footnote changes

Endnote changes

---

## Author Response (AR2)

Response to the comments from Reviewer #1

We greatly appreciate the reviewer for his/her thorough review and constructive comments. We have revised our manuscript as much as possible following his/her comments. Our response to each comment is described as follows:

*1. (l. 195) Matsuno (1971) is not the right reference for this discussion. That paper describes a modeling, not observational, study and it does not show the quadrupole structure.*

> We have added Labitzke (1972) to the reference and revised the sentence. Labitzke (1972) examined temperature profiles from rocketsonde and rocket grenade observations during sudden stratospheric warming, and found that temperature lowers in the Arctic mesosphere and low-latitude stratosphere while temperature rises in the low-latitude mesosphere during the SSW.

*2. More generally related to the above, you say that the mechanism you propose is not the same as that which occurs in stratwarms. However, the morphology of the response looks quite similar. Likewise, both occur during the midwinter months. Could you say more about why you have decided that they are different?*

> We have revised Sect. 4.1. as follows:
> "Matsuno (1971) showed that the quadrupolar structure of temperature change can be interpreted as the transient response to the forcing of planetary waves during stratospheric sudden warming. However, the quadrupolar structure observed in the present study may not be fully explained by the transient response to the planetary waves since the present results are based on the JJA-averaged field."

*3. (l. 195-204) Likewise, the Haynes et al (1991) reference for cross-equatorial effects of downward control is not appropriate. All of the steady state model cases shown in that paper are confined to a single hemisphere and indicate that the circulation response is confined to latitudes close to the wave source. Can you find any modeling support for your interpreted near-global response to steady-state forcing?*

> Semeniuk and Shepherd (2001) examined the middle-atmosphere Hadley circulation and its interaction with extratropical wave-driven circulation, using a numerical model. They showed that the upwelling in low latitudes and downwelling at high latitudes are induced by the wave forcing in the extratropics. This point have already mentioned in the last paragraph of Section 3. To clarify this point, we have revised the sentence (l. 195) 
[revised manuscript text omitted]

difference_6-7.1.docx

| Page 7: Deleted | Yuki Matsushita | 2/6/2020 11:27:00 AM |
|---|---|---|

anomaly

| Page 7: Inserted | Yuki Matsushita | 2/6/2020 11:27:00 AM |
|---|---|---|

change

| Page 7: Deleted | Yuki Matsushita | 2/6/2020 11:27:00 AM |
|---|---|---|

caused by planetary wave breaking

| Page 7: Deleted | Yuki Matsushita | 2/6/2020 11:27:00 AM |
|---|---|---|

in the NH (

| Page 7: Inserted | Yuki Matsushita | 2/6/2020 11:27:00 AM |
|---|---|---|

(Labitzke, 1972).

| Page 7: Deleted | Yuki Matsushita | 2/6/2020 11:27:00 AM |
|---|---|---|

,

| Page 7: Inserted | Yuki Matsushita | 2/6/2020 11:27:00 AM |
|---|---|---|

 (

| Page 7: Deleted | Yuki Matsushita | 2/6/2020 11:27:00 AM |
|---|---|---|

). Since

| Page 7: Inserted | Yuki Matsushita | 2/6/2020 11:27:00 AM |
|---|---|---|

) showed that

| Page 7: Deleted | Yuki Matsushita | 2/6/2020 11:27:00 AM |
|---|---|---|

is observed in the seasonal mean field, the quadrupolar pattern

| Page 7: Deleted | Yuki Matsushita | 2/6/2020 11:27:00 AM |
|---|---|---|

the

| Page 7: Deleted | Yuki Matsushita | 2/6/2020 11:27:00 AM |
|---|---|---|

anomaly shown in the

| Page 7: Inserted | Yuki Matsushita | 2/6/2020 11:27:00 AM |
|---|---|---|

change can be interpreted as the transient response to the forcing of planetary waves during stratospheric sudden warming. However, the quadrupolar structure observed in the

| Page 7: Deleted | Yuki Matsushita | 2/6/2020 11:27:00 AM |
|---|---|---|

is

| Page 7: Inserted | Yuki Matsushita | 2/6/2020 11:27:00 AM |
|---|---|---|

may

| Page 7: Inserted | Yuki Matsushita | 2/6/2020 11:27:00 AM |
|---|---|---|

be fully explained by the transient response to

| Page 7: Deleted | Yuki Matsushita | 2/6/2020 11:27:00 AM |
|---|---|---|

same as the mechanism shown by Matsuno (1971).

| Page 7: Inserted | Yuki Matsushita | 2/6/2020 11:27:00 AM |
|---|---|---|

planetary waves since the present results are based on the JJA-averaged field.

| Page 7: Deleted | Yuki Matsushita | 2/6/2020 11:27:00 AM |
|---|---|---|

of the temperature anomalies shown by

| Page 7: Inserted | Yuki Matsushita | 2/6/2020 11:27:00 AM |
|---|---|---|

in

| Page 7: Deleted | Yuki Matsushita | 2/6/2020 11:27:00 AM |
|---|---|---|

can

| Page 7: Inserted | Yuki Matsushita | 2/6/2020 11:27:00 AM |
|---|---|---|

may

| Page 7: Deleted | Yuki Matsushita | 2/6/2020 11:27:00 AM |
|---|---|---|

anomaly of the

| Page 7: Deleted | Yuki Matsushita | 2/6/2020 11:27:00 AM |
|---|---|---|

circulation with upwelling at low latitudes and

| Page 7: Inserted | Yuki Matsushita | 2/6/2020 11:27:00 AM |
|---|---|---|

flow anomaly around Region A and the

| Page 7: Inserted | Yuki Matsushita | 2/6/2020 11:27:00 AM |
|---|---|---|

anomaly

| Page 7: Inserted | Yuki Matsushita | 2/6/2020 11:27:00 AM |
|---|---|---|

below Region A

| Page 7: Inserted | Yuki Matsushita | 2/6/2020 11:27:00 AM |
|---|---|---|

extratropical

| Page 7: Deleted | Yuki Matsushita | 2/6/2020 11:27:00 AM |
|---|---|---|

below through

| Page 7: Inserted | Yuki Matsushita | 2/6/2020 11:27:00 AM |

from

| Page 7: Deleted | Yuki Matsushita | 2/6/2020 11:27:00 AM |

principle

| Page 7: Inserted | Yuki Matsushita | 2/6/2020 11:27:00 AM |

theory

| Page 7: Inserted | Yuki Matsushita | 2/6/2020 11:27:00 AM |

Since the $\overline{M}_y$ around the equator at ~40 km is small in the strong wave forcing years (Fig. 5), this poleward flow anomaly extends to low latitudes and crosses the equator at the altitude of ~40 km (Fig.6). And then, upwelling anomaly is formed at low latitudes due to the mass continuity.

| Page 11: Inserted | Yuki Matsushita | 2/6/2020 11:27:00 AM |

Haynes, P. H., McIntyre, M. E., Shepherd, T. G., Marks, C. J. and Shine, K. P.: On the "Downward Control" of Extratropical Diabatic Circulations by Eddy-Induced Mean Zonal Forces, J. Atmos. Sci., 48(4), 651–678, doi:10.1175/1520-0469(1991)048<0651:OTCOED>2.0.CO;2, 1991.

| Page 12: Inserted | Yuki Matsushita | 2/6/2020 11:27:00 AM |

Labitzke, K.: The Interaction Between Stratosphere and Meosphere in Winter, J. Atmos. Sci., 29(7), 1395–1399, doi:10.1175/1520-0469(1972)029<1395:TIBSAM>2.0.CO;2, 1972.

Header and footer changes

Text Box changes

Header and footer text box changes

Footnote changes

Endnote changes